# LOSS LANDSCAPE MATTERS: TRAINING CERTIFIABLY ROBUST MODELS WITH FAVORABLE LOSS LANDSCAPE

## ABSTRACT

In this paper, we study the problem of training certifiably robust models. Certifiable training minimizes an upper bound on the worst-case loss over the allowed perturbation, and thus the tightness of the upper bound is an important factor in building certifiably robust models. However, many studies have shown that Interval Bound Propagation (IBP) training uses much looser bounds but outperforms other models that use tighter bounds. We identify another key factor that influences the performance of certifiable training: *smoothness of the loss landscape*. We consider linear relaxation-based methods and find significant differences in the loss landscape across these methods. Based on this analysis, we propose a certifiable training method that utilizes a tighter upper bound and has a landscape with favorable properties. The proposed method achieves performance comparable to state-of-the-art methods under a wide range of perturbations.

## 1 INTRODUCTION

Despite the success of deep learning in many applications, the existence of adversarial example, an imperceptibly modified input that is designed to fool the neural network (Szegedy et al., 2013; Biggio et al., 2013), hinders the application of deep learning to safety-critical domains. There has been increasing interest in building a model that is robust to adversarial attacks (Goodfellow et al., 2014; Papernot et al., 2016; Kurakin et al., 2016; Madry et al., 2018; Tramèr et al., 2017; Zhang et al., 2019a; Xie et al., 2019). However, most defense methods evaluate their robustness with adversarial accuracy against predefined attacks such as PGD attack (Madry et al., 2018) or C&W attack (Carlini & Wagner, 2017). Thus, these defenses can be broken by new attacks (Athalye et al., 2018).

To this end, many training methods have been proposed to build a certifiably robust model that can be guaranteed to be robust to adversarial perturbations (Hein & Andriushchenko, 2017; Raghunathan et al., 2018b; Wong & Kolter, 2018; Dvijotham et al., 2018; Mirman et al., 2018; Gowal et al., 2018; Zhang et al., 2019b). They develop an upper bound on the worst-case loss over valid adversarial perturbations and minimize it to train a certifiably robust model. These certifiable training methods can be mainly categorized into two types: linear relaxation-based methods and bound propagation methods. Linear relaxation-based methods use relatively tighter bounds, but are slow, hard to scale to large models, and memory-inefficient (Wong & Kolter, 2018; Wong et al., 2018; Dvijotham et al., 2018). On the other hand, bound propagation methods, represented by Interval Bound Propagation (IBP), are fast and scalable due to the use of simple but much looser bounds (Mirman et al., 2018; Gowal et al., 2018). One would expect that training with tighter bounds would lead to better performance, but IBP outperforms linear relaxation-based methods in many cases, despite using much looser bounds.

These observations on the performance of certifiable training methods raise the following questions:

*Why does training with tighter bounds not result in a better performance? What other factors may influence the performance of certifiable training? How can we improve the performance of certifiable training methods with tighter bounds?*

In this paper, we provide empirical and theoretical analysis to answer these questions. First, we demonstrate that IBP (Gowal et al., 2018) has a more favorable loss landscape than other linear

relaxation-based methods, and thus it often leads to better performance even with much looser bounds. To account for this difference, we present a unified view of IBP and linear relaxation-based methods and find that the relaxed gradient approximation (which will be defined in Definition 1) of each method plays a crucial role in its optimization behavior. Based on the analysis of the loss landscape and the optimization behavior, we propose a new certifiable training method that has a favorable landscape with tighter bounds. The performance of the proposed method is comparable to that of state-of-the-art methods under a wide range of perturbations. We summarize the contributions of this study as follows:

- We provide empirical and theoretical analysis of the loss landscape of certifiable training methods and find that smoothness of the loss landscape is important for building certifiably robust models.

- We propose a certifiable training method with tighter bounds and a favorable loss landscape, obtaining comparable performance with state-of-the-art methods under a wide range of perturbations.

## 2 RELATED WORK

Earlier studies on training certifiably robust models were limited to 2-layered networks (Hein & Andriushchenko, 2017; Raghunathan et al., 2018a). To scale to larger networks, a line of work has proposed the use of linear relaxation of nonlinear activation to formulate a robust optimization. Then, a dual problem is considered and a dual feasible solution is used to simplify the computation further. By doing so, Wong & Kolter (2018) built a method that can scale to a 4-layered network, and later, Wong et al. (2018) used Cauchy random projections to scale to much larger networks. However, they are still slow and memory-inefficient. Dvijotham et al. (2018) proposed a method called predictor-verifier training (PVT), which uses a verifier network to optimize the dual solution. This is similar to our proposed method but we do not require any additional network. Xiao et al. (2018) proposed to add regularization technique with adversarial training for inducing ReLU stability, but it is less effective than other certified defenses. We also encourage our model to avoid unstable ReLUs, but we train the model with an upper bound of the worst-case loss and investigate ReLU stability from the loss landscape perspective.

Mirman et al. (2018) proposed the propagation of a geometric bound (called domain) through the network to yield an outer approximation in logit space. This can be done with an efficient layerwise computation that exploits interval arithmetic. Over the outer domain, one can compute the worst-case loss to be minimized during training. Gowal et al. (2018) used a special case of the domain propagation called Interval Bound Propagation (IBP) using the simplest domain, the interval domain (or interval bound). In IBP, the authors introduced a different objective function, heuristic scheduling on the hyperparameters, and elision of the last layer to stabilize the training and to improve the performance.

Both approaches, linear relaxation-based methods and bound propagation methods, use an upper bound on the worst-case loss. Bound propagation methods exploit much looser upper bounds, but they enjoy an unexpected benefit in many cases: better robustness than linear relaxation-based methods. Balunovic & Vechev (2019) hypothesized that the complexity of the loss computation makes the optimization more difficult, which could be a reason why IBP outperforms linear relaxation-based methods. They proposed a new optimization procedure with the existing linear relaxation. In this paper, we further investigate the causes of the difficulties in the optimization. Recently, Zhang et al. (2019b) proposed CROWN-IBP which uses linear relaxation in a verification method called CROWN (Zhang et al., 2018) in conjunction with IBP to train a certifiably robust model.

Although beyond our focus here, there is another line of work on randomized smoothing (Li et al., 2018; Lecuyer et al., 2019; Cohen et al., 2019; Salman et al., 2019), which can probabilistically certify the robustness with arbitrarily high probability by using a smoothed classifier. However, it requires a large number of samples for inference.

There are many other works on certifiable verification (Weng et al., 2018; Singh et al., 2018a; 2019; 2018b; Zhang et al., 2018; Boopathy et al., 2019; Lyu et al., 2020). However, our work focuses on "certifiable training".

# 3 BACKGROUND

First, we provide a brief overview of certifiable training methods. Then, we consider IBP (Gowal et al., 2018) as a special case of linear relaxation-based methods. This unified view on certifiable training methods helps us to comprehensively analyze the differences between the two approaches: bound propagation and linear relaxation. We present the details of the IBP in Appendix B.

## 3.1 NOTATIONS AND CERTIFIABLE TRAINING

We consider a $c$-class classification problem with a neural network $f(\boldsymbol{x}; \boldsymbol{\theta})$ with the layerwise operations $\boldsymbol{z}^{(k)} = h^{(k)}(\boldsymbol{z}^{(k-1)})$ ($k = 1, \cdots, K$) and the input $\boldsymbol{z}^{(0)} = \boldsymbol{x}$ in the input space $\mathcal{X}$. The corresponding probability function is denoted by $\boldsymbol{p}_f = \text{softmax} \circ f : \mathcal{X} \rightarrow [0, 1]^c$ with subscript $f$. We denote a subnetwork with $k$ operations as $h^{[k]} = h^{(k)} \circ \cdots \circ h^{(1)}$. For a linear operation $h^{(k)}$, we use $\boldsymbol{W}^{(k)}$ and $\boldsymbol{b}^{(k)}$ to denote the weight and the bias for the layer. We consider the robustness of the classifier against the norm-bounded perturbation set $\mathbb{B}(\boldsymbol{x}, \epsilon) = \{\boldsymbol{x}' \in \mathcal{X} : ||\boldsymbol{x}' - \boldsymbol{x}|| \leq \epsilon\}$ with the perturbation level $\epsilon$. Here, we mainly focus on the $\ell_\infty$-norm bounded set. To compute the margin between the true class $y$ for the input $\boldsymbol{x}$ and the other classes, we define a $c \times c$ matrix $\boldsymbol{C}(y) = \boldsymbol{I} - \boldsymbol{1}e^{(y)^T}$ with $(\boldsymbol{C}(y)\boldsymbol{z}^{(K)})_m = \boldsymbol{z}_m^{(K)} - \boldsymbol{z}_y^{(K)}$ ($m = 0, \cdots, c-1$). For the last linear layer, the weights $\boldsymbol{W}^{(K)}$ and the bias $\boldsymbol{b}^{(K)}$ are merged with $\boldsymbol{C}(y)$, that is, $\boldsymbol{W}^{(K)} \equiv \boldsymbol{C}(y)\boldsymbol{W}^{(K)}$ and $\boldsymbol{b}^{(K)} \equiv \boldsymbol{C}(y)\boldsymbol{b}^{(K)}$, yielding the margin score function $\boldsymbol{s}(\boldsymbol{x}, y; \boldsymbol{\theta}) = \boldsymbol{C}(y)f(\boldsymbol{x}; \boldsymbol{\theta}) = f(\boldsymbol{x}; \boldsymbol{\theta}) - f_y(\boldsymbol{x}; \boldsymbol{\theta})\boldsymbol{1}$ satisfying $\boldsymbol{p_s} = \boldsymbol{p_f}$. Then we can define the worst-case margin score $\boldsymbol{s}^*(\boldsymbol{x}, y, \epsilon; \boldsymbol{\theta}) = \max_{\boldsymbol{x}' \in \mathbb{B}(\boldsymbol{x}, \epsilon)} \boldsymbol{s}(\boldsymbol{x}', y; \boldsymbol{\theta})$ where $\max$ is element-wise maximization. With an upper bound $\overline{\boldsymbol{s}}$ on the worst-case margin score, $\overline{\boldsymbol{s}} \geq \boldsymbol{s}^*$, we can provide an upper bound on the worst-case loss over valid adversarial perturbations as follows:

$$\mathcal{L}(\overline{\boldsymbol{s}}(\boldsymbol{x}, y, \epsilon; \boldsymbol{\theta}), y) \geq \max_{\boldsymbol{x}' \in \mathbb{B}(\boldsymbol{x}, \epsilon)} \mathcal{L}(f(\boldsymbol{x}'; \boldsymbol{\theta}), y) \tag{1}$$

for cross-entropy loss $\mathcal{L}$ (Wong & Kolter, 2018). Therefore, we can formulate certifiable training as a minimization of the upper bound, $\min_{\boldsymbol{\theta}} \mathcal{L}(\overline{\boldsymbol{s}}(\boldsymbol{x}, y, \epsilon; \theta), y)$, instead of directly solving $\min_{\boldsymbol{\theta}} \max_{\boldsymbol{x}' \in \mathbb{B}(\boldsymbol{x}, \epsilon)} \mathcal{L}(f(\boldsymbol{x}'; \theta), y)$ which is infeasible. Note that adversarial training (Madry et al., 2018) uses a strong iterative gradient-based attack (PGD) to provide a lower bound on the worst-case loss to be minimized, but it cannot provide a certifiably robust model. Whenever possible, we will simplify the notations by omitting variables such as $\boldsymbol{x}, y, \epsilon$, and $\boldsymbol{\theta}$.

## 3.2 LINEAR RELAXATION-BASED METHODS

For a subnetwork $h^{[k]}$, given with the pre-activation upper/lower bounds, $\boldsymbol{u}$ and $\boldsymbol{l}$, for each nonlinear activation function $h$ in $h^{[k]}$, linear relaxation-based methods (Wong & Kolter (2018); Wong et al. (2018); Zhang et al. (2019b)) use a relaxation of the activation function by two elementwise linear function bounds, $\overline{h}$ and $\underline{h}$, that is, $\underline{h}(\boldsymbol{z}) \leq h(\boldsymbol{z}) \leq \overline{h}(\boldsymbol{z})$ for $\boldsymbol{l} \leq \boldsymbol{z} \leq \boldsymbol{u}$. We denote the function bounds as $\overline{h}(\boldsymbol{z}) = \overline{\boldsymbol{a}} \odot \boldsymbol{z} + \overline{\boldsymbol{b}}$ and $\underline{h}(\boldsymbol{z}) = \underline{\boldsymbol{a}} \odot \boldsymbol{z} + \underline{\boldsymbol{b}}$ for some $\overline{\boldsymbol{a}}, \overline{\boldsymbol{b}}, \underline{\boldsymbol{a}}$, and $\underline{\boldsymbol{b}}$, where $\odot$ denotes the elementwise (Hadamard) product. Using all the function bounds $\overline{h}$'s and $\underline{h}$'s for the nonlinear activations in conjunction with the linear operations in $h^{[k]}$, an $i$th (scalar) activation $h_i^{[k]}(\cdot) \in \mathbb{R}$ can be upper bounded by a linear function $\boldsymbol{g}^T \cdot + b$ over $\mathbb{B}(\boldsymbol{x}, \epsilon)$ as in Zhang et al. (2018). This can be equivalently explained with the dual relaxation viewpoint in Wong & Kolter (2018). Further details are provided in Appendix C. Now we are ready to upper bound the activation $h_i^{[k]}$ over $\mathbb{B}(\boldsymbol{x}, \epsilon)$.

**Definition 1** (Linear Relaxation with Relaxed Gradient Approximation). *For each neuron activation $h_i^{[k]}$, a linear relaxation method computes an upper approximation of the activation over $\mathbb{B}(\boldsymbol{x}, \epsilon)$ by using $\boldsymbol{g} \in \mathbb{R}^d$ and $b \in \mathbb{R}$ as follows:*

$$\max_{\boldsymbol{x}' \in \mathbb{B}(\boldsymbol{x}, \epsilon)} h_i^{[k]}(\boldsymbol{x}') \leq \max_{\boldsymbol{x}' \in \mathbb{B}(\boldsymbol{x}, \epsilon)} \boldsymbol{g}^T \boldsymbol{x}' + b = \boldsymbol{g}^T \boldsymbol{x} + \epsilon ||\boldsymbol{g}||_* + b. \tag{2}$$

*We call $\boldsymbol{g}$ the relaxed gradient approximation of $h_i^{[k]}$ over $\mathbb{B}(\boldsymbol{x}, \epsilon)$.*

Similarly, we can obtain the corresponding lower bound. Inductively using these upper/lower bounds on the output of the subnetwork, we can obtain the bounds for the next subnetwork $h^{[k+1]}$ and then for the whole network $\boldsymbol{s}$. The final bound $\overline{\boldsymbol{s}}$ on the whole network $\boldsymbol{s}$ can then be used in the objective

(1). The tightness of the bounds $\overline{s}$ and $\mathcal{L}(\overline{s}, y)$ highly depend on how the linear bounds $\overline{h}$ and $\underline{h}$ in each layer are chosen.

**Unified view of IBP and linear relaxation-based methods**  IBP can also be considered as a linear relaxation-based method using zero-slope ($\overline{a} = \underline{a} = 0$) linear bounds, $\overline{h}(z) = u^+$ and $\underline{h}(z) = l^+$, where $v^+ = \max(v, 0)$ and $v^- = \min(v, 0)$. Thus, the bounds of a nonlinear activation depend only on the pre-activation bounds $u$ and $l$ for the activation layer, substantially reducing the feed-forward/backpropagation computations. CROWN-IBP (Zhang et al., 2019b) applies different linear relaxation schemes to the subnetworks and the whole network. It uses the same linear bounds as IBP for the subnetworks $h^{[k]}$ for $k < K$ except for the network $s = h^{[K]}$ itself, and uses $\overline{h}(z) = \frac{u^+}{u^+ - l^-} \odot (z - l^-)$ and $\underline{h}(z) = \mathbf{1}[u^+ + l^- > 0] \odot z$ for the whole network $s$. Moreover, CROWN-IBP uses interpolations between two bounds with the mixing weight $\beta$, IBP bound and CROWN-IBP bound, with the following objective:

$$\mathcal{L}\left((1 - \beta)\overline{s}^{\text{IBP}}(x, y, \epsilon; \theta) + \beta\overline{s}^{\text{CROWN-IBP}}(x, y, \epsilon; \theta), y\right). \quad (3)$$

Convex Adversarial Polytope (CAP) (Wong & Kolter, 2018; Wong et al., 2018) uses the linear bounds $\overline{h}(z) = \frac{u^+}{u^+ - l^-} \odot (z - l^-)$ and $\underline{h}(z) = \frac{u^+}{u^+ - l^-} \odot z$ for all subnetworks $h^{[k]}$ and the entire network. As CAP utilizes the linear bounds for each neuron, it is slow and memory-inefficient. It can be easily shown that tighter relaxations on nonlinear activations yield a tighter bound on the worst-case margin score $s^*$. To specify the linear relaxation variable $\phi \equiv \{\overline{a}, \underline{a}, \overline{b}, \underline{b}\}$ used in relaxation, we use the notation $\overline{s}(x, y, \epsilon; \theta, \phi)$. CROWN-IBP and CAP generally yield a much tighter bound than IBP. These relaxation schemes are illustrated in Figure 6 in Appendix D.

## 4   WHAT FACTORS INFLUENCE THE PERFORMANCE OF CERTIFIABLE TRAINING?

One would expect that a tighter upper bound on the worst-case loss in (1) is beneficial in certifiable training. However, several previous works have shown that this is not the case: IBP performs better than linear relaxation-based methods in many cases while utilizing a much looser bound. We investigate the loss landscape and the optimization behavior of IBP and other linear relaxation-based methods, and find that the non-smoothness of the relaxed gradient approximation of linear relaxations negatively affects their performance. Detailed settings of the following analyses are presented in Appendix A.

### 4.1   LOSS LANDSCAPE OF CERTIFIABLE TRAINING

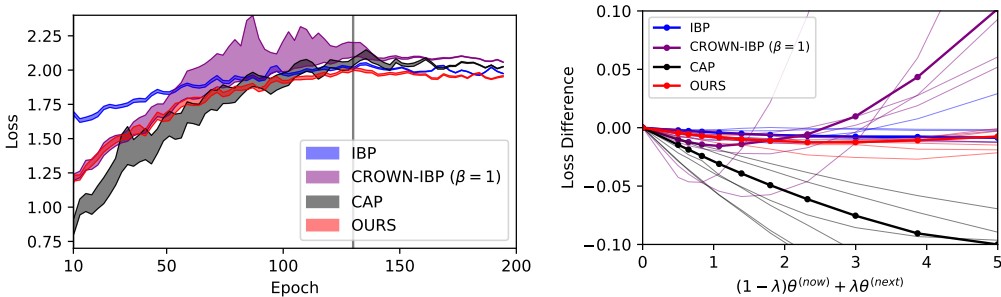

Figure 1: (*Left*) The learning curves for the scheduled value of $\epsilon$ with the loss variation along the gradient descent direction (the vertical line indicates when the ramp-up ends), and (*Right*) the loss landscapes along the gradient descent direction at each training step in the later phase of ramp-up period (epoch 50-130). The thin lines and thick lines in the figure on the right show some sample landscapes at each step and the median values, respectively. Our method shows tight bounds like CROWN-IBP, while its landscape is as favorable as IBP, achieving the best performance among these four methods (see Table 1).

We empirically show that models that have tighter bounds, CROWN-IBP (Zhang et al., 2019b) and CAP (Wong & Kolter, 2018), tend to have non-smooth loss landscapes, which hinder optimization during training. We examine the learning curves of IBP and these linear relaxation-based methods. For a simple analysis, we avoid considering the mixture of the two logits in (3), and use $\beta = 1$ to consider CROWN-IBP logit only. Figure 1 (left) shows the learning curves on CIFAR-10 under $\epsilon_{train} = 8/255$. We use $\epsilon$-scheduling with the warm-up (regular training) for the first 10 epochs and the ramp-up during epochs 10-130 where we linearly increases the perturbation radius from 0 to the target perturbation $\epsilon_{train}$. Thus, the training loss may increase even during learning.

In the early phase of the ramp-up period, in which the models are trained with small $\epsilon$, CAP and CROWN-IBP have lower losses than IBP as expected because they use much tighter relaxation bounds than IBP. In particular, CAP has much tighter bounds than the others because CAP uses tighter relaxations for each subnetwork. This is consistent with the known results, that CAP tends to outperform the others at small perturbations, such as $\epsilon_{train} = 2/255$ on CIFAR-10 (see Table 1 for details). However, at the end of the training, when the perturbation reaches its maximum target value ($\epsilon_{train}$), the opposite result is observed where CAP and CROWN-IBP perform worse than IBP.

To understand this inconsistency, we measure the variation of the loss along the gradient direction as in Santurkar et al. (2018), which is represented as the shaded region in Figure 1 (left). We find that linear relaxation-based methods have large variations, while IBP maintains a small variation throughout the entire training phase. It is known that a smooth loss landscape with a small loss variation induces stable and fast optimization with well-behaved gradients (Santurkar et al., 2018). Therefore, even though CAP and CROWN-IBP show robustness in the early phase of training, the non-smooth loss landscape in the ramp-up period might have hindered the optimization, yielding less robust models. As will be discussed in the following section, we find that the loss variation is highly related to the relaxed gradient approximation $g$ used in linear relaxation.

We further explore the loss landscape near the local region of the parameter space at the current parameter $\theta^{(now)}$ toward the next parameter $\theta^{(next)}$ along the gradient in Figure 1 (right). We plot the landscapes for the later phase of the ramp-up period (epochs 50-130) during which large perturbations are used. IBP has flatter landscapes compared to the others, whereas CROWN-IBP has landscapes with large curvature along the gradient, and thus it tends to move towards a sharp local minimum and it may remain stuck there. Therefore, it may overfit to be robust to small perturbations, but is not robust to the target perturbation $\epsilon_{train}$.

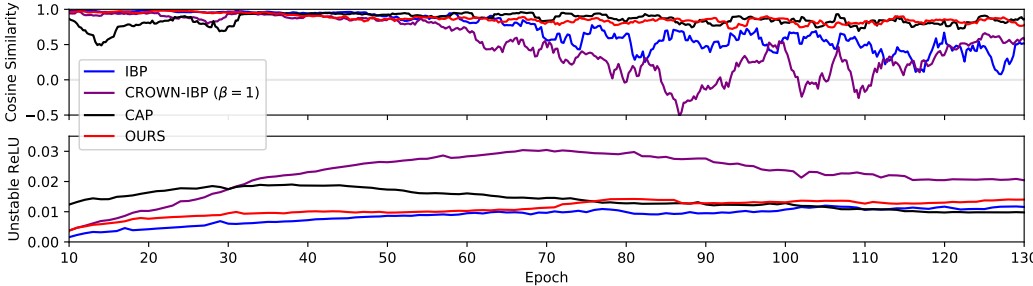

Figure 2: (*Top*) Cosine similarities between two consecutive loss gradients and (*Bottom*) the ratio of the number of unstable ReLUs during the ramp-up period. A large number of unstable ReLUs, high nonlinearity, leads to an unfavorable landscape that can negatively affect the optimization process.

Next, we establish a relationship between the optimization procedure and linear relaxation. Figure 2 (top) shows the directional deviation between two successive loss gradient steps in terms of cosine similarity during training. Simultaneously, Figure 2 (bottom) shows the ratio of the number of unstable ReLUs for which the pre-activation bounds $l$ and $u$ span zero. We observe that the cosine similarity value is low when the number of unstable ReLUs is large - for example, in the early stage of CAP and the middle stage of CROWN-IBP. In particular, in the middle of the ramp-up period, CROWN-IBP has a large number of unstable ReLUs and exhibits abrupt changes in gradient steps. It often has deviation angles larger than $90°$, leading to parameter updates in the opposite direction of the previous one, bouncing in the basin of a local minimum. This is consistent with the results

shown in Figure 1. Moreover, since the gradient directions are not well-aligned, it may not enjoy the advantages of momentum-based optimizers and be sensitive to the learning rate. To summarize, a large number of unstable ReLUs, high nonlinearity, leads to an unfavorable landscape that can negatively affect the optimization process.

## 4.2 SMOOTHNESS OF RELAXED GRADIENT APPROXIMATION

In this section, we investigate the loss landscape further from a theoretical perspective to answer the question: "What makes some landscapes more favorable than others?" We find that the relaxed gradient approximation of a linear relaxation affects the smoothness of the landscape. First, we need some mild smoothness assumptions that are natural when the network parameters $\boldsymbol{\theta}_1$ and $\boldsymbol{\theta}_2$ are close to each other, especially they are two consecutive parameters from SGD update.

**Assumption 1.** *Given linear relaxation method, we make the following assumptions on the bias $b(\boldsymbol{x}; \boldsymbol{\theta})$ in the linear relaxation and the probability function $\boldsymbol{p}(\boldsymbol{x}; \boldsymbol{\theta})$:*
*(1) $||\nabla_{\boldsymbol{\theta}} b(\boldsymbol{x}; \boldsymbol{\theta}_1) - \nabla_{\boldsymbol{\theta}} b(\boldsymbol{x}; \boldsymbol{\theta}_2)|| \leq L_{\boldsymbol{\theta\theta}}^b ||\boldsymbol{\theta}_1 - \boldsymbol{\theta}_2||$ for all $\boldsymbol{\theta}_1, \boldsymbol{\theta}_2$ and $\boldsymbol{x}$.*

*(2) $||\boldsymbol{p}(\boldsymbol{x}; \boldsymbol{\theta}_1) - \boldsymbol{p}(\boldsymbol{x}; \boldsymbol{\theta}_1)|| \leq L_{\boldsymbol{\theta}}^{\boldsymbol{p}} ||\boldsymbol{\theta}_1 - \boldsymbol{\theta}_2||$ for all $\boldsymbol{\theta}_1, \boldsymbol{\theta}_2$ and $\boldsymbol{x}$.*

With the above assumptions, we can provide an upper bound on the loss gradient difference for linear relaxation-based methods to measure the non-smoothness of the loss landscape as follows:

**Theorem 1.** *Given input $\boldsymbol{x} \in \mathcal{X}$ and perturbation radius $\epsilon$, let $M$ be $\max_{\boldsymbol{x}' \in \mathbb{B}(\boldsymbol{x}, \epsilon)} ||\boldsymbol{x}'||$. For a linear relaxation-based method with the upper bound $\overline{\boldsymbol{s}}_m(\boldsymbol{x}; \boldsymbol{\theta}) = \max_{\boldsymbol{x}' \in \mathbb{B}(\boldsymbol{x}, \epsilon)} \boldsymbol{g}^{(m)}(\boldsymbol{x}; \boldsymbol{\theta})^T \boldsymbol{x}' + b^{(m)}(\boldsymbol{x}; \boldsymbol{\theta})$, if $b^{(m)}$ satisfies Assumption 1 (1) for each $m$ and $\boldsymbol{p}_s$ satisfies Assumption 1 (2), then*

$$||\nabla_{\boldsymbol{\theta}} \mathcal{L}(\overline{\boldsymbol{s}}(\boldsymbol{x}; \boldsymbol{\theta}_1)) - \nabla_{\boldsymbol{\theta}} \mathcal{L}(\overline{\boldsymbol{s}}(\boldsymbol{x}; \boldsymbol{\theta}_2))||$$
$$\leq \max_m \left( 2\epsilon ||\nabla_{\boldsymbol{\theta}} \boldsymbol{g}^{(m)}(\boldsymbol{x}; \boldsymbol{\theta}_{1,2})|| + M ||\nabla_{\boldsymbol{\theta}} \boldsymbol{g}^{(m)}(\boldsymbol{x}; \boldsymbol{\theta}_1) - \nabla_{\boldsymbol{\theta}} \boldsymbol{g}^{(m)}(\boldsymbol{x}; \boldsymbol{\theta}_2)|| + L^{(m)} ||\boldsymbol{\theta}_1 - \boldsymbol{\theta}_2|| \right)$$
$$(4)$$

*for any $\boldsymbol{\theta}_1, \boldsymbol{\theta}_2$, where $L^{(m)} = L_{\boldsymbol{\theta\theta}}^{b^{(m)}} + L_{\boldsymbol{\theta}}^{\boldsymbol{p}_s} ||\nabla_{\boldsymbol{\theta}} \overline{\boldsymbol{s}}(\boldsymbol{x}; \boldsymbol{\theta}_{1,2})||$ and $\boldsymbol{\theta}_{1,2}$ can be any of $\boldsymbol{\theta}_1$ and $\boldsymbol{\theta}_2$.*

According to Theorem 1, the relaxed gradient approximations $\boldsymbol{g}^{(m)}$ in the linear relaxation play a major role in shaping the loss landscape. The smoother the relaxed gradient approximations are, the smoother the loss landscape is. Especially for IBP, using the zero-slope relaxed gradient approximation $\boldsymbol{g}^{(m)} \equiv \boldsymbol{0}$ for all $m$, the loss difference is upper bounded by only the last term, $\max L^{(m)} ||\boldsymbol{\theta}_1 - \boldsymbol{\theta}_2||$, and it is relatively small for a single gradient step. On the other hand, for other linear relaxation-based methods using non-zero relaxed gradient approximation $\boldsymbol{g}^{(m)} \neq \boldsymbol{0}$, the gradient updates used in the training are more unstable than IBP. It is consistent with the empirical results shown in Figure 1 that there are significant differences between the loss variations of IBP and others.

## 5 PROPOSED METHOD

Our analyses so far suggest that tightness of the upper bound on the worst-case loss and smoothness of the loss landscape are important for building a certifiably robust model. Therefore, we aim to design a new certifiable training method to improve the aforementioned factors (favorable landscape and tighter bound).

**More favorable landscape via less $\underline{a} = 1$** We observe that CROWN-IBP ($\beta = 1$) tends to have more unstable ReLU and less smooth landscape than the others. What, in the objective of CROWN-IBP, does lead to these results? To answer this question, we investigate variants of CROWN-IBP with different $\underline{a}$ settings for unstable ReLUs. For each setting, we sample $\underline{a} \in \{0, 1\}$ with different $(p, q)$ with $P(\underline{a} = 1 \mid |l| > |u|) = p$ and $P(\underline{a} = 1 \mid |l| \leq |u|) = q$ for each neuron with pre-activation bounds $l$ and $u$. We use $\underline{a} = \mathbf{1}[\boldsymbol{u}^+ + \boldsymbol{l}^- > \boldsymbol{0}]$ for the other stable ReLUs. For the other elements of the linear relaxation variable $\boldsymbol{\phi} = \{(\overline{\boldsymbol{a}}, \underline{\boldsymbol{a}}, \overline{\boldsymbol{b}}, \underline{\boldsymbol{b}})\}$, we fix $\overline{\boldsymbol{a}} = \frac{\boldsymbol{u}^+}{\boldsymbol{u}^+ - \boldsymbol{l}^-}$, $\overline{\boldsymbol{b}} = -\frac{\boldsymbol{u}^+ \boldsymbol{l}^-}{\boldsymbol{u}^+ - \boldsymbol{l}^-}$, and $\underline{\boldsymbol{b}} = \boldsymbol{0}$ for each activation node, because they are the optimal choices for tightening the bound (see Appendix C.2 for details). Figure 3 shows that it tends to have more unstable ReLUs as the number

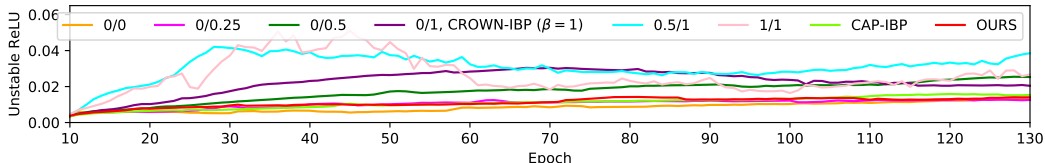

Figure 3: The ratio of the number of unstable ReLUs for models with different $\underline{a}$ settings during the ramp-up period. Notation $p/q$ denotes the variant with sampling $\underline{a} \in \{0, 1\}$ with $P(\underline{a} = 1 \mid |l| > |u|) = p$ and $P(\underline{a} = 1 \mid |l| \leq |u|) = q$ for unstable ReLUs. As the number of $\underline{a} = 1$ increases, it tends to have more unstable ReLUs, which leads to less smooth loss landscapes.

of $\underline{a}$ satisfying $\underline{a} = 1$ increases. This observation implies that it is required to have smaller portion of $\underline{a}$ with $\underline{a} = 1$ to have a more favorable landscape.

However, reducing the portion of $\underline{a}$ with $\underline{a} = 1$ is not enough to achieve robustness unless the tightness is guaranteed. Through manually adjusting the $\underline{a}$, variants of CROWN-IBP achieve favorable landscapes, but they show looser upper bounds which lead to a worse performance. Further investigation of variants of CROWN-IBP is presented in Appendix E. Therefore, it is required to search for appropriate values of $\underline{a}$ that can achieve both tightness and favorable landscape.

**Tighter bound via optimization** Now, we aim to reduce the number of $\underline{a}$ satisfying $\underline{a} = 1$ and to tighten the upper bound in (1), simultaneously. We can achieve both by minimizing the upper bound over the linear relaxation variable $\phi$ as follows:

$$\mathcal{L}(\overline{s}(x, y, \epsilon; \theta), y) \geq \min_{\phi} \mathcal{L}(\overline{s}(x, y, \epsilon; \theta, \phi), y) \geq \max_{x' \in \mathbb{B}(x, \epsilon)} \mathcal{L}(f(x'; \theta), y). \tag{5}$$

It can be equivalently understood as solving the dual optimization in CAP rather than using a dual feasible solution. However, solving the dual optimization is computationally prohibited for the linear relaxation of CAP. To resolve this problem, we use the same linear relaxation as IBP for the subnetworks of $s$ except for $s$ itself, similar to CROWN-IBP. Further, we efficiently compute a surrogate $\hat{\underline{a}}$ of the minimizer $\underline{a}^* = \arg\min_{\underline{a}} \mathcal{L}(\overline{s}(x, y, \epsilon; \theta, \phi), y)$ using the one-step projected gradient update of the relaxation variable $\underline{a}$. Specifically, we have

$$\hat{\underline{a}} = \Pi_{[0,1]^n} \left( \underline{a}_0 - \eta \text{sign}(\nabla_{\underline{a}} \mathcal{L}(\overline{s}(x, y, \epsilon; \theta, \phi), y)) \right) \tag{6}$$

with an initial point $\underline{a}_0 \sim U[0,1]^n$ and $\eta \geq 1$, yielding the final objective $\mathcal{L}(\overline{s}(x, y, \epsilon; \theta, \hat{\phi}), y)$ where $\hat{\phi} = \{(\overline{a}, \hat{\underline{a}}, \overline{b}, \underline{b})\}$.

## 6 EXPERIMENTS

In this section, we demonstrate the proposed method satisfies two key criteria required for building certifiably robust models: 1) tightness of the upper bound on the worst-case loss, and 2) smoothness of the loss landscape. Subsequently, we evaluate the performance of the method by comparing it with others certifiable training methods. Details on the experimental settings are in Appendix A.

**Tightness** To validate that the proposed method (OURS) has tighter bounds than other relaxations, we analyze various linear relaxation methods in Figure 4. We define a tightness measure as a sum over the worst-case margin for each class $m$, $\sum_{m=0}^{c-1} \overline{s}_m(x, y, \epsilon; \theta)$, obtained from (2). Then, we evaluate multiple methods on a single fixed model pre-trained with the proposed training method. The compared methods are, from left to right, OURS, CROWN-IBP (Zhang et al., 2019b), CAP-IBP, and RANDOM. All methods use the same IBP relaxation for subnetworks, but use different linear relaxation variables $\underline{a}$ for the whole network $s$. CROWN-IBP, CAP-IBP, and RANDOM use $\underline{a} = \mathbf{1}[u^+ + l^- > 0]$, $\underline{a} = \frac{u^+}{u^+ - l^-}$ and $\underline{a} \sim U[0,1]^n$, respectively. We fix the other variables $\overline{a}, \overline{b}$, and $\underline{b}$, as in Section 5. In both figures, our method shows the lowest value on average, which indicates that a single gradient step in (6) is sufficient to obtain tighter bounds compared to other relaxation methods. See Appendix P for the equivalent tightness violin plots of other models.

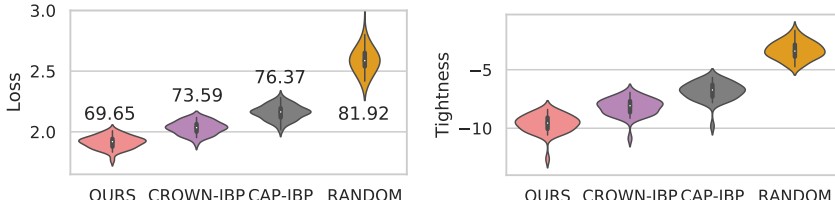

Figure 4: Violin plots of the test loss with the corresponding verified error (*Left*) and of tightness (*Right*) for various linear relaxations. Lower is better. This shows that the proposed relaxation method has a tighter bound than the others relaxation methods.

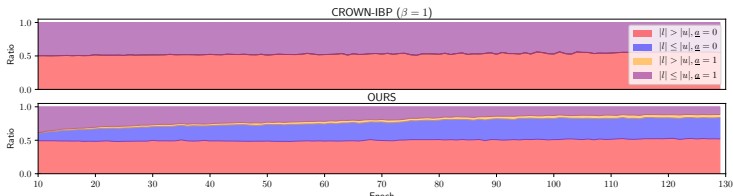

Figure 5: The ratio of number of $\underline{a}$ used for unstable ReLUs during the ramp-up period. Note that the blue region denotes the number of $\underline{a} = 0$ when $|l| \leq |u|$. It indicates that the proposed method reduces the number of $\underline{a}$ satisfying $\underline{a} = 1$ (purple+yellow).

**Smoothness** Figure 1 shows that the proposed method has small loss variations along the gradient as with IBP, whereas CROWN-IBP ($\beta = 1$) has a wide range of loss values. This is because CROWN-IBP ($\beta = 1$) has more unstable ReLUs than our methods as shown in Figure 2. As mentioned above, number of $\underline{a}$ is closely related to the amount of unstable ReLUs, and Figure 5 shows that our method has successfully reduced the number of $\underline{a} = 1$. Further, we conduct analysis on smoothness of the loss landscape with the loss gradient change (the left term in (4)) in Appendix H.

**Robustness** We evaluate the performance of the proposed method and compare it to that of state-of-the-art certifiable training methods: IBP (Gowal et al., 2018), CROWN-IBP ($\beta = 1$) (Zhang et al., 2019b), and CAP (Wong et al., 2018), as in Section 4.1. On MNIST, we follow Zhang et al. (2019b) and use $\epsilon_{train} \geq \epsilon_{test}$; whereas for CAP, we use the same $\epsilon_{train} = \epsilon_{test}$ which yields better results. We used three evaluation metrics: standard (clean) error, 100-step PGD error, and verified error. For the verified error, we evaluated with the bound $\overline{s}$ of each method.

Table 1: Test errors (Standard / PGD / Verified error) of IBP, CROWN-IBP ($\beta = 1$), CAP, and OURS on MNIST, CIFAR-10, and SVHN. Bold and underline numbers are the first and second lowest verified error.

| Data | $\epsilon(l_\infty)$ | IBP | CROWN-IBP ($\beta = 1$) | CAP | OURS |
|---|---|---|---|---|---|
| **MNIST** | $\epsilon = 0.1$ | 1.18 / 2.16 / 3.52 | 1.07 / 1.69 / **2.10** | 0.80 / 1.73 / 3.19 | 1.09 / 1.77 / 2.36 |
| | $\epsilon = 0.2$ | 2.00 / 3.29 / 6.31 | 2.99 / 5.50 / 7.97 | 3.22 / 6.72 / 11.06 | 1.70 / 3.44 / **4.34** |
| | $\epsilon = 0.3$ | 3.50 / 5.85 / 10.45 | 5.73 / 10.76 / 16.28 | 19.19 / 35.84 / 47.85 | 3.49 / 5.59 / **9.79** |
| | $\epsilon = 0.4$ | 3.50 / 7.30 / 17.96 | 5.73 / 14.63 / 23.80 | - | 3.49 / 6.77 / **15.42** |
| **CIFAR 10** | $\epsilon = 2/255$ | 37.98 / 49.40 / 55.39 | 32.48 / 42.77 / 50.15 | 28.8 / 38.95 / **48.50** | 31.49 / 42.73 / 49.42 |
| | $\epsilon = 4/255$ | 46.42 / 57.42 / 62.80 | 45.56 / 58.24 / 64.47 | 40.78 / 52.62 / 61.88 | 42.53 / 55.55 / **61.52** |
| | $\epsilon = 6/255$ | 52.84 / 63.92 / 68.79 | 54.72 / 65.28 / 71.04 | 49.20 / 60.85 / 69.03 | 50.19 / 61.88 / **66.90** |
| | $\epsilon = 8/255$ | 55.71 / 66.79 / 70.95 | 61.37 / 70.66 / 75.37 | 56.77 / 66.78 / 73.02 | 56.01 / 66.17 / **69.70** |
| | $\epsilon = 16/255$ | 67.10 / 75.12 / 78.26 | 76.65 / 81.90 / 84.42 | 75.11 / 80.67 / 82.07 | 65.93 / 75.39 / **77.87** |
| **SVHN** | $\epsilon = 0.01$ | 19.91 / 34.12 / 43.83 | 17.25 / 30.84 / 39.88 | 16.88 / 30.16 / **37.09** | 16.41 / 30.43 / 39.44 |

Table 1 summarizes the evaluation results under different $\epsilon_{test}$ for each dataset. In general, when $\epsilon_{test}$ is low, methods with tighter linear relaxations show good performance, whereas IBP tends to perform better as $\epsilon_{test}$ increases. In short, the state-of-the-art methods perform well for a specific range of $\epsilon_{test}$. For example, IBP show relatively better performance in the case of $\epsilon_{test} = 0.3, 0.4$ on MNIST and $\epsilon_{test} = 6/255, 8/255, 16/255$ on CIFAR-10. On the other hand, CAP and CROWN-IBP ($\beta = 1$) outperform IBP in the case of $\epsilon_{test} = 0.1$ on MNIST, $\epsilon_{test} = 2/255$ on CIFAR-10 and $\epsilon_{test} = 0.001$ on SVHN. This result is consistent with the analysis shown in Figure 1 that CAP and CROWN-IBP ($\beta = 1$) have lower loss than IBP at small $\epsilon$, but their loss landscape is less smooth than IBP, leading to worse performance at large $\epsilon$. Moreover, CAP cannot be trained on MNIST when $\epsilon_{train} = 0.4$. As the case is also not specified in Wong et al. (2018), it seems that CAP is hard to be robust to $\epsilon_{train} \geq 0.4$. On the other hand, the proposed method shows consistent performance in a wide range of $\epsilon_{test}$ values, achieving the best performance in most cases, since it has tighter bounds and a favorable landscape, not overfitting to a local minimum during the $\epsilon$-scheduling. We also compared our method with other prior work (Xiao et al., 2018; Mirman et al., 2018; Balunovic & Vechev, 2019) in Appendix K. We also conduct additional experiments on the hyperparameters in Appendix L, M, and N.

Unlike standard training, certifiable training requires $\epsilon$-scheduling. It is implicitly assumed that a set of weights that makes the network robust to a small $\epsilon$ is a good initial point to learn robustness to a large $\epsilon_{train}$. However, linear relaxation-based methods with tighter bounds start with a lower loss at a small $\epsilon$, but with an unfavorable loss landscape, they cannot explore a sufficiently large area of the parameter space. Hence, they overfit to be robust to a small perturbation, and not generalize to a large perturbation. CAP and CROWN-IBP ($\beta = 1$) are typical examples that demonstrate the overfitting. This may overregularize the weight norm and decrease the model capacity (Wong et al., 2018; Zhang et al., 2019b). The tightness of the proposed method improves the performance for a small $\epsilon$, while the smoothness of the proposed method helps the optimization process, which also leads to better performance for a large $\epsilon$. To conclude, the proposed method can achieve a decent performance under a wide range of perturbations as shown in Table 1.

Table 2: Test errors (Standard / PGD / Verified error) of OURS and CROWN-IBP$_{1 \to 0}$ on CIFAR-10. Bold numbers are the lower error.

| $\epsilon(\ell_\infty)$ | $\epsilon = 2/255$ | $\epsilon = 4/255$ | $\epsilon = 6/255$ | $\epsilon = 8/255$ | $\epsilon = 16/255$ |
|---|---|---|---|---|---|
| **OURS** | **31.49 / 42.73 / 49.42** | **42.53 / 55.55 / 61.52** | **50.19 / 61.88 / 66.90** | 56.01 / **66.17** / **69.70** | **65.93 / 75.39 / 77.87** |
| **CROWN-IBP**$_{1 \to 0}$ | 36.30 / 47.13 / 52.70 | 45.92 / 57.58 / 62.07 | 53.54 / 64.14 / 67.35 | **55.09** / 66.68 / 69.97 | 66.62 / 76.13 / 77.88 |

**Understanding $\beta$-scheduling** For CROWN-IBP, we use two different settings of $\beta$ in (7), CROWN-IBP$_{1 \to 1}$ and CROWN-IBP$_{1 \to 0}$, where the subscript $\beta_{start} \to \beta_{end}$ refers to the linear scheduling on $\beta$ from $\beta_{start}$ to $\beta_{end}$. Zhang et al. (2019b) found that the $\beta$-scheduling of CROWN-IBP$_{1 \to 0}$ could help to improve the robustness performance. And they argued that this is because training with a tighter bound of CROWN-IBP at the beginning can provide a good initialization for later IBP training. On the other hand, we provide another explanation that CROWN-IBP$_{1 \to 0}$ starts with a tighter bound (CROWN-IBP only) but not overfits to small perturbation by gradually introducing the IBP objective which has a smoother landscape. Despite using a single objective without the mixture parameter $\beta$, the proposed method can outperforms CROWN-IBP$_{1 \to 0}$ on CIFAR-10 as shown in Table 2.

## 7 CONCLUSION

In this work, we have investigated the loss landscape of certifiable training and found that the smoothness of the loss landscape is an important factor that influences in building certifiably robust models. To this end, we proposed a method that satisfies the two criteria: tightness of the upper bound on the worst-case loss and smoothness of the loss landscape. Then, we empirically demonstrated that the proposed method achieves robustness comparable to state-of-the-art methods under a wide range of perturbations. We believe that with an improved understanding of the loss landscape, better certifiably robust models can be built.

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

## A  EXPERIMENTAL SETTINGS

**Datasets and Architectures**   In the experiments, we use three datasets: MNIST, CIFAR-10 and
SVHN and model architectures (Small, Medium, and Large) in Gowal et al. (2018) and their vari-
ants (Small* and Large*) as follows:

- Small: Conv($\cdot$,16,4,2) - Conv(16,32,4,1) - Flatten - FC($\cdot$,100) - FC(100,c)

- Small*: Conv($\cdot$,16,4,2) - Conv(16,32,4,2) - Flatten - FC($\cdot$,100) - FC(100,c)

- Medium: Conv($\cdot$,32,3,1) - Conv(32,32,4,2) - Conv(32,64,3,1) - Conv(64,64,4,2) - Flatten -
  FC($\cdot$,512) - FC(512,512) - FC(512,c)

- Large:  Conv($\cdot$,64,3,1)  -  Conv(64,64,3,1)  -  Conv(64,128,3,2)  -  Conv(128,128,3,1)  -
  Conv(128,128,3,1) - Flatten - FC($\cdot$,512) - FC(512,c)

- Large*: Conv($\cdot$,64,3,1) - Conv(64,128,3,2) - Conv(128,128,3,1) - Conv(128,128,3,1) - Flat-
  ten - FC($\cdot$,512) - FC(512,c)

where Conv($c_1, c_2, k, s$) is a conv layer with input channel $c_1$, output channel $c_2$, kerner size $k$, and
stride $s$, and FC($d_1, d_2$) is a fully-connected layer with input dimension $d_1$ and output dimension $d_2$.
All layers are followed by ReLU activation except for the last layer and the flatten layer (Flatten).

**Loss and training schedules**   For general training schedules, we refer to Appendix C, D of Zhang
et al. (2019b) with a single GPU (Titan Xp). We use the following mixed cross-entropy loss as in
Zhang et al. (2019b):

$$\kappa \mathcal{L}\left(f(\boldsymbol{x}; \boldsymbol{\theta}), y\right) + (1 - \kappa)\mathcal{L}\left((1 - \beta)\overline{s}^{\text{IBP}}(\boldsymbol{x}, y, \epsilon; \boldsymbol{\theta}) + \beta \overline{s}^{\text{MODEL}}(\boldsymbol{x}, y, \epsilon; \boldsymbol{\theta}), y\right), \tag{7}$$

where $\kappa$ is the mixing weight between the natural loss and the robust loss, and $\beta$ is the mixing weight
between the two bounds obtained with IBP and given relaxation method (e.g. CROWN-IBP).

### A.1  SETTINGS IN SECTION 4.1

**Figure 1**   We conduct the experiment in Figure 1 on CIFAR-10 dataset with Medium architecture
over all four methods. We train the model with $\epsilon_{train} = {}^8/_{255}$ for 200 epochs using $\epsilon$-scheduling
with 10 warm-up epochs and 120 ramp-up epochs. We use Adam optimizer with learning rate 0.001.
We reduce the learning rate by 50% every 10 epochs after $\epsilon$-scheduling ends.

To demonstrate the instability of each training, we describe the variation of the loss along the gra-
dient direction as Santurkar et al. (2018). We take steps of different lengths in the direction of the
gradient and measure the loss values obtained at each step. For the sake of consistency, we fix a
Cauchy random matrix when evaluating CAP to obtain deterministic loss landscapes, not introduc-
ing randomness. The loss variation is computed with

$$\mathcal{L}(\overline{s}(\boldsymbol{\theta}(t)))$$
$$\text{where } \mathcal{L}(\overline{s}(\boldsymbol{\theta})) \equiv \mathcal{L}(\overline{s}(\boldsymbol{x}, y, \epsilon; \boldsymbol{\theta}), y) \text{ and}$$
$$\boldsymbol{\theta}(t) \equiv \boldsymbol{\theta}_0 - t\eta \nabla_{\boldsymbol{\theta}} \mathcal{L}(\overline{s}(\boldsymbol{\theta}_0)) \text{ for } t \in [0, 5], \tag{8}$$

where $\boldsymbol{\theta}_0 (= \boldsymbol{\theta}(0))$ is the current model parameters and $\eta$ is the learning rate. For the step of length
$t$, we sample ten points from a range of [0,5] on a log scale. In Figure 1 (right), $\boldsymbol{\theta}^{(now)} = \boldsymbol{\theta}(0)$ and
$\boldsymbol{\theta}^{(next)} = \boldsymbol{\theta}(1)$.

**Figure 2 (top)**   In Figure 2, with the same model used in Figure 1, we plot cosine similarity be-
tween two successive loss gradient steps during training as follows:

$$cos(\nabla_{\boldsymbol{\theta}} \mathcal{L}(\overline{s}(\boldsymbol{\theta}(0))), \nabla_{\boldsymbol{\theta}} \mathcal{L}(\overline{s}(\boldsymbol{\theta}(1)))),$$

where $cos(\boldsymbol{v}_1, \boldsymbol{v}_2)$ is the cosine value of the angle between two vectors $\boldsymbol{v}_1$ and $\boldsymbol{v}_2$ .

## A.2 SETTINGS IN TABLE 1

For MNIST, we use the same hyper-parameters as in Appendix C of Zhang et al. (2019b). We train for 200 epochs (10 warm-up epochs and 50 ramp-up epochs) on Large model with batch sizes of 100. we decay the learning rate, 0.0005, by 10% in [130,190] epochs. As mentioned in Zhang et al. (2019b), we also found the same issue when training with small $\epsilon$ (see Appendix N for details). To alleviate the issue, we use $\epsilon_{train} = \min(0.4, \epsilon_{test} + 0.1)$ for each $\epsilon_{test}$ as Table 2 of Zhang et al. (2019b).

For CIFAR-10, we train for 400 epochs (20 warm-up epochs and 240 ramp-up epochs) on Medium model with batch sizes of 128. We decay the learning rate, 0.003, by $2\times$ every 10 epochs after the ramp-up period.

For SVHN, we train for 200 epochs (10 warm-up epochs and 120 ramp-up epochs) on Large model with batch sizes of 128 (OURS with batch sizes of 80 to avoid out of memory). We decay the learning rate, 0.0003, by $2\times$ every 10 epochs after the ramp-up period. Only for SVHN, we apply normalization with mean (0.438, 0.444, 0.473) and standard deviation (0.198, 0.201, 0.197) for each channel.

In Table 1, we use $\kappa$-scheduling from 1 to 0. For the corresponding results of $\kappa$-scheduling from 0 to 0, we refer the reader to Table 5.

We modify the source code for CAP[1] to match our settings. For example, we introduce the warm-up period and linear $\epsilon$-scheduling. We avoid using the reported results in the literature and aim to make a fair comparison under the same settings with only minor differences - for example, because CAP does not support the channel-wise normalization, we could not use the input normalization. Also, due to the memory limit of CAP, we use a smaller batch size of 32 and try other smaller architectures. We found that it often achieves better results with smaller architectures (similar to the results in Table 3 of Wong et al. (2018)). Thus, we present the performance with Large*, Medium, and Small* on MNIST, CIFAR-10, and SVHN, respectively. Throughout the experiments, CAP uses the fixed $\kappa = 0$.

## B  INTERVAL BOUND PROPAGATION (IBP)

IBP (Gowal et al., 2018) starts from the interval bound $\mathcal{I}^{(0)} \equiv \{\boldsymbol{z} : \boldsymbol{l}^{(0)} \leq \boldsymbol{z} \leq \boldsymbol{u}^{(0)}\} = \mathbb{B}(\boldsymbol{x}, \epsilon)$ in the input space with the upper bound $\boldsymbol{u}^{(0)} = \boldsymbol{x} + \epsilon\mathbf{1}$ and the lower bound $\boldsymbol{l}^{(0)} = \boldsymbol{x} - \epsilon\mathbf{1}$ where $\mathbf{1}$ is a column vector filled with 1. Then we propagate the interval bound $\mathcal{I}^{(k-1)} \equiv \{\boldsymbol{z} : \boldsymbol{l}^{(k-1)} \leq \boldsymbol{z} \leq \boldsymbol{u}^{(k-1)}\}$ by using following equations iteratively:

$$\boldsymbol{u}^{(k)} = h^{(k)}(\boldsymbol{u}^{(k-1)}) \text{ and } \boldsymbol{l}^{(k)} = h^{(k)}(\boldsymbol{l}^{(k-1)}) \tag{9}$$

for element-wise monotonic increasing nonlinear activation $h^{(k)}$ with the pre-activation bounds $\boldsymbol{u}^{(k-1)}$ and $\boldsymbol{l}^{(k-1)}$, and

$$\boldsymbol{u}^{(k)} = \boldsymbol{W}^{(k)}\left(\frac{\boldsymbol{u}^{(k-1)} + \boldsymbol{l}^{(k-1)}}{2}\right) + |\boldsymbol{W}^{(k)}|\left(\frac{\boldsymbol{u}^{(k-1)} - \boldsymbol{l}^{(k-1)}}{2}\right) + \boldsymbol{b}^{(k)} \text{ and} \tag{10}$$

$$\boldsymbol{l}^{(k)} = \boldsymbol{W}^{(k)}\left(\frac{\boldsymbol{u}^{(k-1)} + \boldsymbol{l}^{(k-1)}}{2}\right) - |\boldsymbol{W}^{(k)}|\left(\frac{\boldsymbol{u}^{(k-1)} - \boldsymbol{l}^{(k-1)}}{2}\right) + \boldsymbol{b}^{(k)} \tag{11}$$

for linear function $h^{(k)}$ ($k = 1, \cdots, K$). Finally, IBP uses the worst-case margin $\overline{\boldsymbol{s}} = \boldsymbol{u}^{(K)}$ to formulate the objective in (1) for certifiable training.

---

[1] https://github.com/locuslab/convex_adversarial

## C   DETAILS ON LINEAR RELAXATION

### C.1   LINEAR RELAXATION EXPLAINED IN CROWN (ZHANG ET AL., 2018)

To make the paper self-contained, we provide details of linear relaxation given in the supplementary material of CROWN (Zhang et al., 2018). We refer readers to the supplementary for more details. Given a network $h^{[k]}$, we want to upper bound the activation $h_i^{[k]}$. We have $h_i^{[k]}(\boldsymbol{x}') = \boldsymbol{W}_{i,:}^{(k)} h^{(k-1)}(h^{[k-2]}(\boldsymbol{x}')) + \boldsymbol{b}_i^{(k)} = \boldsymbol{W}_{i,:}^{(k)} h^{(k-1)}(\boldsymbol{z}^{(k-2)'}) + \boldsymbol{b}_i^{(k)}$ where $\boldsymbol{z}^{(k-2)'} = h^{[k-2]}(\boldsymbol{x}')$. With the linear function bounds of $\overline{h}^{(k-1)}$ and $\underline{h}^{(k-1)}$ on the activation function $h^{(k-1)}$, we have

$$
\begin{aligned}
h_i^{[k]}(\boldsymbol{x}') &= \boldsymbol{W}_{i,:}^{(k)} h^{(k-1)}(\boldsymbol{z}^{(k-2)'}) + \boldsymbol{b}_i^{(k)} \\
&\leq \sum_{\boldsymbol{W}_{i,j}^{(k)}<0} \boldsymbol{W}_{i,j}^{(k)} \underline{h}_j^{(k-1)}(\boldsymbol{z}^{(k-2)'}) + \sum_{\boldsymbol{W}_{i,j}^{(k)}\geq0} \boldsymbol{W}_{i,j}^{(k)} \overline{h}_j^{(k-1)}(\boldsymbol{z}^{(k-2)'}) + \boldsymbol{b}_i^{(k)} \\
&= \sum_{\boldsymbol{W}_{i,j}^{(k)}<0} \boldsymbol{W}_{i,j}^{(k)} \underline{a}_j^{(k-1)} \boldsymbol{z}_j^{(k-2)'} + \sum_{\boldsymbol{W}_{i,j}^{(k)}\geq0} \boldsymbol{W}_{i,j}^{(k)} \overline{a}_j^{(k-1)} \boldsymbol{z}_j^{(k-2)'} \\
&\quad + \sum_{\boldsymbol{W}_{i,j}^{(k)}<0} \boldsymbol{W}_{i,j}^{(k)} \underline{b}_j^{(k-1)} + \sum_{\boldsymbol{W}_{i,j}^{(k)}\geq0} \boldsymbol{W}_{i,j}^{(k)} \overline{b}_j^{(k-1)} + \boldsymbol{b}_i^{(k)} \\
&= \tilde{\boldsymbol{W}}_{i,:}^{(k)} \boldsymbol{z}^{(k-2)'} + \tilde{\boldsymbol{b}}_i^{(k)} \\
&= \tilde{\boldsymbol{W}}_{i,:}^{(k)} h^{[k-2]}(\boldsymbol{x}') + \tilde{\boldsymbol{b}}_i^{(k)} \\
&= \tilde{\boldsymbol{W}}_{i,:}^{(k)} \left( \boldsymbol{W}^{(k-2)}(h^{[k-3]}(\boldsymbol{x}')) + \boldsymbol{b}^{(k-2)} \right) + \tilde{\boldsymbol{b}}_i^{(k)} \\
&= \hat{\boldsymbol{W}}_{i,:}^{(k-2)} h^{(k-3)}(\boldsymbol{z}^{(k-3)'}) + \hat{\boldsymbol{b}}_i^{(k-2)},
\end{aligned}
$$

where $\tilde{\boldsymbol{W}}_{i,:}^{(k)} = \boldsymbol{W}_{i,:}^{(k)} \boldsymbol{D}^{(k-1)}$ with the diagonal matrix $\boldsymbol{D}_{j,j}^{(k-1)} = \underline{a}_j^{(k-1)}$ for $j$ satisfying $\boldsymbol{W}_{i,j}^{(k)} < 0$ and $\boldsymbol{D}_{j,j}^{(k-1)} = \overline{a}_j^{(k-1)}$ for $j$ satisfying $\boldsymbol{W}_{i,j}^{(k)} \geq 0$, $\tilde{\boldsymbol{b}}_i^{(k)} = \sum_{\boldsymbol{W}_{i,j}^{(k)}<0} \boldsymbol{W}_{i,j}^{(k)} \underline{b}_j^{(k-1)} + \sum_{\boldsymbol{W}_{i,j}^{(k)}\geq0} \boldsymbol{W}_{i,j}^{(k)} \overline{b}_j^{(k-1)} + \boldsymbol{b}_i^{(k)}$, $\hat{\boldsymbol{W}}_{i,:}^{(k-2)} = \tilde{\boldsymbol{W}}_{i,:}^{(k)} \boldsymbol{W}^{(k-2)}$, and $\hat{\boldsymbol{b}}_i^{(k-2)} = \tilde{\boldsymbol{W}}_{i,:}^{(k)} \boldsymbol{b}^{(k-2)} + \tilde{\boldsymbol{b}}_i^{(k)}$. Applying similar method iteratively, we can obtain $\boldsymbol{g}$ and $b$ in (2) for the linear relaxation of $h_i^{[k]}$.

### C.2   DUAL OPTIMIZATION VIEW

We first modify some notations in the main paper and use the notations similar to Wong & Kolter (2018). We use the following hat notations: $\hat{\boldsymbol{z}}^{(k+1)} = \boldsymbol{W}^{(k+1)} \boldsymbol{z}^{(k)} + \boldsymbol{b}^{(k+1)}$ and $\boldsymbol{z}^{(k)} = h^{(k)}(\hat{\boldsymbol{z}}^{(k)})$ where $h^{(k)}$ is the $k$-th nonlinear activation function. We can build a primal problem with $\boldsymbol{c}^T = \boldsymbol{C}_{m,:}$ as follows:

$$
\max_{\boldsymbol{z}^{(K)}} \boldsymbol{c}^T \hat{\boldsymbol{z}}^{(K)} \tag{12}
$$

such that

$$
\begin{aligned}
\boldsymbol{x} - \epsilon \boldsymbol{1} &\leq \boldsymbol{z}^{(0)}, \\
\boldsymbol{z}^{(0)} &\leq \boldsymbol{x} + \epsilon \boldsymbol{1}, \\
\hat{\boldsymbol{z}}^{(k+1)} &= \boldsymbol{W}^{(k+1)} \boldsymbol{z}^{(k)} + \boldsymbol{b}^{(k+1)} \ (k = 0, \cdots, K-1), \text{ and} \\
\boldsymbol{z}^{(k)} &= h^{(k)}(\hat{\boldsymbol{z}}^{(k)}) \ (k = 1, \cdots, K-1).
\end{aligned}
$$

Note that our $c$ is negation of that of Wong & Kolter (2018). Now we can derive the dual of the primal (12) as follows:

$$
\min_{\substack{\boldsymbol{\xi}^+, \boldsymbol{\xi}^- \geq \mathbf{0} \\ \boldsymbol{\nu}_k}} \sup_{\boldsymbol{z}^{(k)}, \hat{\boldsymbol{z}}^{(k)}} \boldsymbol{c}^T \boldsymbol{z}^{(\hat{K})} + \boldsymbol{\xi}^{-T}(\boldsymbol{x} - \epsilon \mathbf{1} - \boldsymbol{z}^{(0)}) + \boldsymbol{\xi}^{+T}(\boldsymbol{z}^{(0)} - \boldsymbol{x} - \epsilon \mathbf{1})
$$

$$
+ \sum_{k=0}^{K-1} \boldsymbol{\nu}_{k+1}^T \left( \hat{\boldsymbol{z}}^{(k+1)} - (\boldsymbol{W}^{(k+1)} \boldsymbol{z}^{(k)} + \boldsymbol{b}^{(k+1)}) \right) + \sum_{k=1}^{K-1} \hat{\boldsymbol{\nu}}_k^T \left( \boldsymbol{z}^{(k)} - h^{(k)}(\hat{\boldsymbol{z}}^{(k)}) \right)
$$

$$
= (\boldsymbol{c} + \boldsymbol{\nu}_K)^T \hat{\boldsymbol{z}}^{(K)} + (\boldsymbol{\xi}^+ - \boldsymbol{\xi}^- - \boldsymbol{W}^{(1)T} \boldsymbol{\nu}_1)^T \boldsymbol{z}^{(0)} + \sum_{k=1}^{K-1} (-\boldsymbol{W}^{(k+1)T} \boldsymbol{\nu}_{k+1} + \hat{\boldsymbol{\nu}}_k)^T \boldsymbol{z}^{(k)}
$$

$$
+ \sum_{k=1}^{K-1} (\hat{\boldsymbol{\nu}}_k^T h^{(k)}(\hat{\boldsymbol{z}}^{(k)}) - \boldsymbol{\nu}_k^T \hat{\boldsymbol{z}}^{(k)}) \tag{13}
$$

$$
- \boldsymbol{\nu}_1^T \boldsymbol{b}^{(1)} - \boldsymbol{\xi}^T \boldsymbol{x} - \epsilon \|\boldsymbol{\xi}\|_1.
$$

It leads to $\boldsymbol{c} + \boldsymbol{\nu}_K = \mathbf{0}, \boldsymbol{\xi}^+ - \boldsymbol{\xi}^- - \boldsymbol{W}^{(1)T} \boldsymbol{\nu}_1 = \mathbf{0}$, and $-\boldsymbol{W}^{(k+1)T} \boldsymbol{\nu}_{k+1} + \hat{\boldsymbol{\nu}}_k = \mathbf{0} \ (k = 1, \cdots, K-1)$. Alternatively, they are represented as follows:

$$
\boldsymbol{\nu}_K = -\boldsymbol{c},
$$
$$
\hat{\boldsymbol{\nu}}_k = \boldsymbol{W}^{(k+1)T} \boldsymbol{\nu}_{k+1} \ (k = K-1, \cdots, 1), \text{ and}
$$
$$
\boldsymbol{\xi} = \hat{\boldsymbol{\nu}}_1.
$$

Now we need relationship between $\hat{\boldsymbol{\nu}}_k$ and $\boldsymbol{\nu}_k$, i.e., $\boldsymbol{\nu}_k = g(\hat{\boldsymbol{\nu}}_k)$. With the further relaxation $\boldsymbol{\nu}_k = \boldsymbol{\alpha}_k \odot \hat{\boldsymbol{\nu}}_k$, we have a relaxed problem as follows:

$$
\min_{\boldsymbol{\alpha}_k} \sup_{\boldsymbol{z}^{(k)}, \hat{\boldsymbol{z}}^{(k)}} \sum_{k=1}^{K-1} (\hat{\boldsymbol{\nu}}_k^T h^{(k)}(\hat{\boldsymbol{z}}^{(k)}) - \boldsymbol{\nu}_k^T \hat{\boldsymbol{z}}^{(k)}) - \boldsymbol{\nu}_1^T \boldsymbol{b}^{(1)} - \boldsymbol{\xi}^T \boldsymbol{x} - \epsilon \|\boldsymbol{\xi}\|_1 \tag{14}
$$

such that

$$
\boldsymbol{\nu}_K = -\boldsymbol{c},
$$
$$
\hat{\boldsymbol{\nu}}_k = \boldsymbol{W}^{(k+1)T} \boldsymbol{\nu}_{k+1} \ (k = K-1, \cdots, 1),
$$
$$
\boldsymbol{\nu}_k = \boldsymbol{\alpha}_k \odot \hat{\boldsymbol{\nu}}_k \ (k = K-1, \cdots, 1), \text{ and}
$$
$$
\boldsymbol{\xi} = \hat{\boldsymbol{\nu}}_1.
$$

We decompose the first term in (14), and ignore the subscript $k$ as follows $\hat{\boldsymbol{\nu}}^T h(\hat{\boldsymbol{z}}) - (\boldsymbol{\alpha} \odot \hat{\boldsymbol{\nu}})^T \hat{\boldsymbol{z}}$. Further, we decompose this for each element, $\hat{\nu} h(\hat{z}) - \alpha \hat{\nu} \hat{z} = \hat{\nu}(h(\hat{z}) - \alpha \hat{z})$. If the pre-activation bounds for $h$ are both positive (active ReLU), then $\alpha$ should be 1 not to make the inner supremum $\infty$. Similarly, if the pre-activation bounds for $h$ are both negative (dead ReLU), then $\alpha$ should be 0. In the case of unstable ReLU ($l \leq 0 \leq u$), if $\hat{\nu} < 0$, then we need to solve $\max_\alpha \inf_{\hat{z}} h(\hat{z}) - \alpha \hat{z}$. The inner infimum is 0 for $0 \leq \alpha \leq 1$, and is $-\infty$ otherwise. On the other hand, if $\hat{\nu} \geq 0$, then we need to solve $\min_\alpha \sup_{\hat{z}} h(\hat{z}) - \alpha \hat{z}$. The inner supremum is $\max\{u - \alpha u, -\alpha l\}$, and thus the optimal dual variable is $\alpha^* = \frac{u}{u-l}$ which yields the optimal value (multiplied by $\hat{\nu}$) as $\hat{\nu}(u - \frac{u}{u-l} u) = -\frac{ul}{u-l} \hat{\nu}$ which is equivalent to using linear relaxation with $\overline{\boldsymbol{a}} \odot \boldsymbol{z} + \overline{b} = \frac{\boldsymbol{u}}{\boldsymbol{u}-\boldsymbol{l}} \odot (\boldsymbol{z} - \boldsymbol{l})$. We can represent it as $\overline{\boldsymbol{a}} \odot \boldsymbol{z} + \overline{b} = \frac{\boldsymbol{u}^+}{\boldsymbol{u}^+ - \boldsymbol{l}^-} \odot (\boldsymbol{z} - \boldsymbol{l}^-)$ to include the case of active/dead ReLU. For the lower linear bound $\underline{h}(\boldsymbol{z}) = \underline{\boldsymbol{a}} \odot \boldsymbol{z} + \underline{\boldsymbol{b}}$ in case of unstable ReLU, we can use any $\mathbf{0} \leq \underline{\boldsymbol{a}} \leq \mathbf{1}$ and $\underline{\boldsymbol{b}} = \mathbf{0}$ according to the dual relaxation with $\boldsymbol{\alpha}$. While CAP and CROWN-IBP use a dual feasible solution like $\boldsymbol{\alpha} = \frac{\boldsymbol{u}^+}{\boldsymbol{u}^+ - \boldsymbol{l}^-}$ or $\boldsymbol{\alpha} = \mathbf{1}[\boldsymbol{u}^+ + \boldsymbol{l}^- > 0]$, our proposed method aims to optimize over the dual variable $\boldsymbol{\alpha}$ or equivalently optimize over $\mathbf{0} \leq \underline{\boldsymbol{a}} \leq \mathbf{1}$ to further tighten the upper bound on the loss.

# D    ILLUSTRATION OF LINEAR RELAXATIONS

Figure 6 provides some illustrations of linear relaxations used in IBP, CAP, CROWN-IBP, and the proposed method. CROWN-IBP adaptively chooses the relaxation variable so that the area between $\overline{h}$ and $\underline{h}$ is minimized. However, the smaller area does not necessarily imply the tighter bound, and the proposed method achieves tighter bounds than CROWN-IBP relaxation as shown in Figure 4.

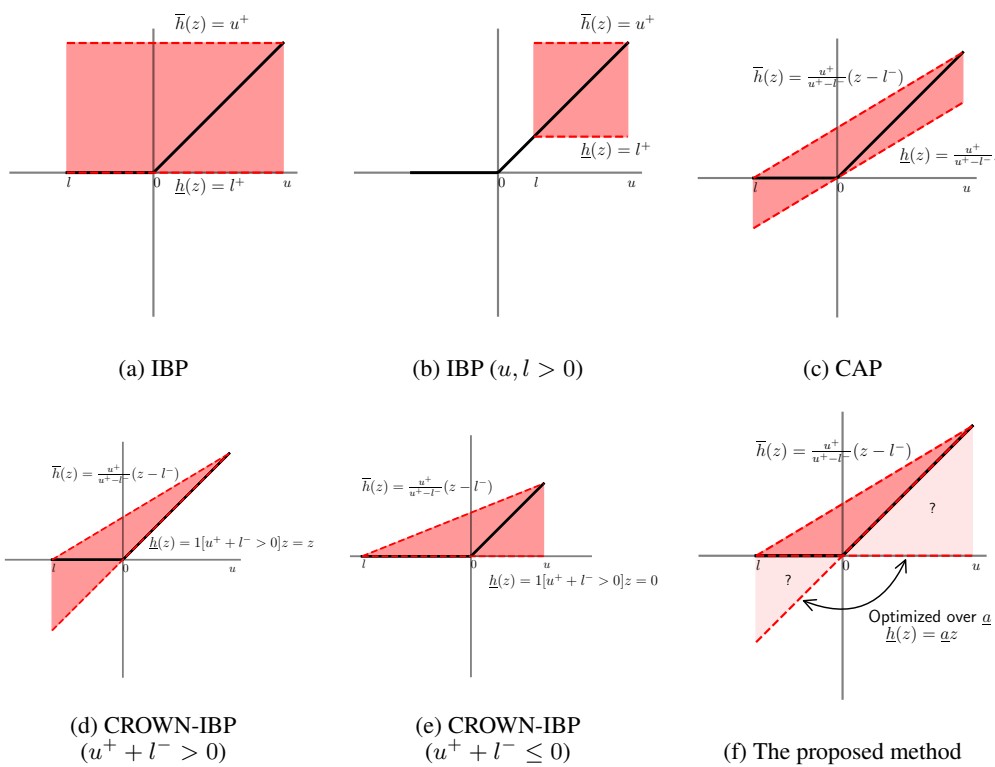

(a) IBP

(b) IBP $(u, l > 0)$

(c) CAP

(d) CROWN-IBP
$(u^+ + l^- > 0)$

(e) CROWN-IBP
$(u^+ + l^- \leq 0)$

(f) The proposed method

Figure 6: Illustrations of linear relaxation methods. Except for (b), they illustrate the relaxations when $l \leq 0 \leq u$ (Unstable ReLU). (b) Illustration of the relaxation of IBP when $u, l > 0$ (Active ReLU).

# E   LEARNING CURVES FOR VARIANTS OF CROWN-IBP

It seems that a certifiable training with a looser bound tends to favor stable ReLUs. For example, IBP starts with small number of unstable ReLUs while CAP starts with large number of ReLUs as shown in Figure 2 (bottom). However, a tighter bound does not directly lead to many unstable ReLUs. We find that $0.5/1$ and $1/1$ have looser bounds than CROWN-IBP (as shown in Figure 7) but they have more unstable ReLUs (as shown in Figure 3) where $p/q$ denotes the variant with sampling $\underline{a} \in \{0, 1\}$ with $P(\underline{a} = 1 \mid |l| > |u|) = p$ and $P(\underline{a} = 1 \mid |l| \leq |u|) = q$ for unstable ReLUs. On the other hand, $0/0$, $0/0.25$, and $0/0.5$ have looser bounds than CROWN-IBP and they have less unstable ReLUs, which leads to small loss variations as in Figure 7. Therefore, this observation implies that it is more important to have less $\underline{a} = 1$ to have a more smooth landscape.

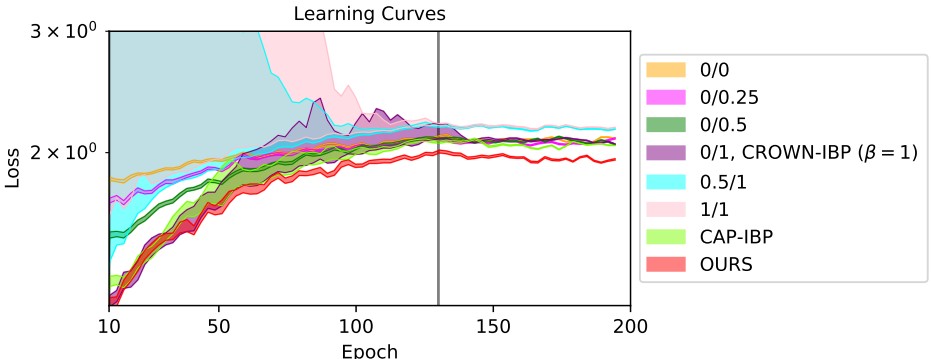

Figure 7: The learning curves for the scheduled value of $\epsilon$ with the loss variation along gradient descent direction (equivalent to Figure 1). As the ratio of the number of $\underline{a}$ with $\underline{a} = 1$ increases, the loss variation increases.

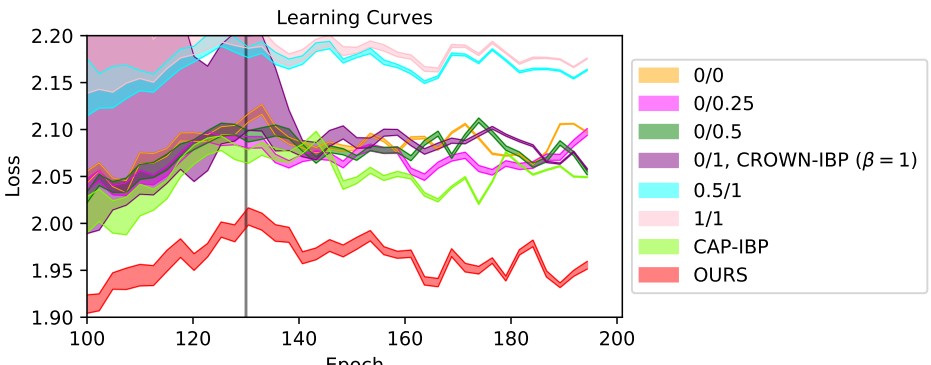

Figure 8: A zoomed-in version of Figure 7 for epochs 100-200.

Table 3: Performance (in terms of errors) of the variants of CROWN-IBP ($\beta = 1$). Note that $0/0.25$, $0/0.5$, and CAP-IBP start with looser bounds but they have more smooth landscape, which leads to a better performance than CROWN-IBP ($\beta = 1$) (highlighted with underline).

| Model | 0/0 | 0/0.25 | 0/0.5 | 0/1 CROWN-IBP ($\beta = 1$) | 0.5/1 | 1/1 | CAP-IBP | OURS |
|---|---|---|---|---|---|---|---|---|
| Standard | 70.66 | 64.50 | 62.72 | 63.24 | 70.69 | 71.41 | 60.36 | 57.14 |
| PGD | 73.84 | 72.67 | 71.42 | 71.70 | 76.68 | 77.03 | 69.46 | 66.88 |
| Verified | 77.60 | 74.47 | 74.92 | 75.72 | 78.38 | 78.73 | 74.29 | **71.45** |

## F PROOF

To prove Theorem 1, we first prove the following proposition. We note that $\boldsymbol{\theta}$ and $\boldsymbol{g}$ are vectorized and the matrix norm of Jacobian is naturally defined - for example, $||\nabla_{\boldsymbol{\theta}}\boldsymbol{g}||$ is induced by the vector norms defined in $\mathcal{X}$ and $\Theta$.

**Proposition 1.** *Given input $\boldsymbol{x} \in \mathcal{X}$ and perturbation radius $\epsilon$, let $M = \max\{||\boldsymbol{x}'|| : \boldsymbol{x}' \in \mathbb{B}(\boldsymbol{x}, \epsilon)\}$. Then, for the upper bound $\overline{s}(\boldsymbol{x}; \boldsymbol{\theta}) = \max_{\boldsymbol{x}' \in \mathbb{B}(\boldsymbol{x}, \epsilon)} \boldsymbol{g}(\boldsymbol{x}; \boldsymbol{\theta})^T \boldsymbol{x}' + b(\boldsymbol{x}; \boldsymbol{\theta})$ with $b$ satisfying Assumption 1 (1), we have*

$$||\nabla_{\boldsymbol{\theta}}\overline{s}(\boldsymbol{x}; \boldsymbol{\theta}_1) - \nabla_{\boldsymbol{\theta}}\overline{s}(\boldsymbol{x}; \boldsymbol{\theta}_2)||$$
$$\leq 2\epsilon||\nabla_{\boldsymbol{\theta}}\boldsymbol{g}(\boldsymbol{x}; \boldsymbol{\theta}_{1,2})|| + M||\nabla_{\boldsymbol{\theta}}\boldsymbol{g}(\boldsymbol{x}; \boldsymbol{\theta}_1) - \nabla_{\boldsymbol{\theta}}\boldsymbol{g}(\boldsymbol{x}; \boldsymbol{\theta}_2)|| + L^b_{\boldsymbol{\theta}\boldsymbol{\theta}}||\boldsymbol{\theta}_1 - \boldsymbol{\theta}_2|| \quad (15)$$

*for any $\boldsymbol{\theta}_1, \boldsymbol{\theta}_2$, where $\boldsymbol{\theta}_{1,2}$ can be any of $\boldsymbol{\theta}_1$ and $\boldsymbol{\theta}_2$.*

*Proof.* Say $\overline{f}(\boldsymbol{x}'; \boldsymbol{\theta}) = \boldsymbol{g}(\boldsymbol{x}; \boldsymbol{\theta})^T \boldsymbol{x}' + b(\boldsymbol{x}; \boldsymbol{\theta})$ and the maximizer $\boldsymbol{x}_i^* = \arg\max_{\boldsymbol{x}' \in \mathbb{B}(\boldsymbol{x}, \epsilon)} \overline{f}(\boldsymbol{x}'; \boldsymbol{\theta}_i)$ for each $\boldsymbol{\theta}_i = \boldsymbol{\theta}_1, \boldsymbol{\theta}_2$. Then, we have

$$||\nabla_{\boldsymbol{\theta}}\overline{s}(\boldsymbol{x}; \boldsymbol{\theta}_1) - \nabla_{\boldsymbol{\theta}}\overline{s}(\boldsymbol{x}; \boldsymbol{\theta}_2)|| = ||\nabla_{\boldsymbol{\theta}}\overline{f}(\boldsymbol{x}_1^*; \boldsymbol{\theta}_1) - \nabla_{\boldsymbol{\theta}}\overline{f}(\boldsymbol{x}_2^*; \boldsymbol{\theta}_2)||$$
$$= ||\nabla_{\boldsymbol{\theta}}\overline{f}(\boldsymbol{x}_1^*; \boldsymbol{\theta}_1) - \nabla_{\boldsymbol{\theta}}\overline{f}(\boldsymbol{x}_2^*; \boldsymbol{\theta}_1) + \nabla_{\boldsymbol{\theta}}\overline{f}(\boldsymbol{x}_2^*; \boldsymbol{\theta}_1) - \nabla_{\boldsymbol{\theta}}\overline{f}(\boldsymbol{x}_2^*; \boldsymbol{\theta}_2)||$$
$$\leq ||\nabla_{\boldsymbol{\theta}}\overline{f}(\boldsymbol{x}_1^*; \boldsymbol{\theta}_1) - \nabla_{\boldsymbol{\theta}}\overline{f}(\boldsymbol{x}_2^*; \boldsymbol{\theta}_1)|| + ||\nabla_{\boldsymbol{\theta}}\overline{f}(\boldsymbol{x}_2^*; \boldsymbol{\theta}_1) - \nabla_{\boldsymbol{\theta}}\overline{f}(\boldsymbol{x}_2^*; \boldsymbol{\theta}_2)||. \quad (16)$$

The first term on the RHS can be upper bounded as follows:

$$||\nabla_{\boldsymbol{\theta}}\overline{f}(\boldsymbol{x}_1^*; \boldsymbol{\theta}_1) - \nabla_{\boldsymbol{\theta}}\overline{f}(\boldsymbol{x}_2^*; \boldsymbol{\theta}_1)|| = ||\nabla_{\boldsymbol{\theta}}(\tilde{\boldsymbol{g}}_1^T \tilde{\boldsymbol{x}}_1^* - \tilde{\boldsymbol{g}}_1^T \tilde{\boldsymbol{x}}_2^*)||$$
$$= ||\nabla_{\boldsymbol{\theta}}(\boldsymbol{g}_1^T \boldsymbol{x}_1^* - \boldsymbol{g}_1^T \boldsymbol{x}_2^*)||$$
$$= ||\nabla_{\boldsymbol{\theta}}\boldsymbol{g}_1(\boldsymbol{x}_1^* - \boldsymbol{x}_2^*)||$$
$$\leq 2\epsilon||\nabla_{\boldsymbol{\theta}}\boldsymbol{g}_1||,$$

where $\boldsymbol{g}_i = \boldsymbol{g}(\boldsymbol{x}; \boldsymbol{\theta}_i)$, $b_i = b(\boldsymbol{x}; \boldsymbol{\theta}_i)$, $\tilde{\boldsymbol{g}}_i^T = [\boldsymbol{g}_i^T; b_i]$ and $\tilde{\boldsymbol{x}}^T = [\boldsymbol{x}^T; 1]$. And the second term on the RHS can be upper bounded as follows:

$$||\nabla_{\boldsymbol{\theta}}\overline{f}(\boldsymbol{x}_2^*; \boldsymbol{\theta}_1) - \nabla_{\boldsymbol{\theta}}\overline{f}(\boldsymbol{x}_2^*; \boldsymbol{\theta}_2)|| = ||\nabla_{\boldsymbol{\theta}}(\tilde{\boldsymbol{g}}_1^T \tilde{\boldsymbol{x}}_2^* - \tilde{\boldsymbol{g}}_2^T \tilde{\boldsymbol{x}}_2^*)||$$
$$= ||\nabla_{\boldsymbol{\theta}}(\tilde{\boldsymbol{g}}_1 - \tilde{\boldsymbol{g}}_2)\tilde{\boldsymbol{x}}_2^*||$$
$$\leq ||\nabla_{\boldsymbol{\theta}}(\boldsymbol{g}_1 - \boldsymbol{g}_2)||||\boldsymbol{x}_2^*|| + ||\nabla_{\boldsymbol{\theta}}(b_1 - b_2)||$$
$$\leq M||\nabla_{\boldsymbol{\theta}}(\boldsymbol{g}_1 - \boldsymbol{g}_2)|| + L^b_{\boldsymbol{\theta}\boldsymbol{\theta}}||\boldsymbol{\theta}_1 - \boldsymbol{\theta}_2||,$$

Therefore, we obtain

$$||\nabla_{\boldsymbol{\theta}}\overline{s}(\boldsymbol{x}; \boldsymbol{\theta}_1) - \nabla_{\boldsymbol{\theta}}\overline{s}(\boldsymbol{x}; \boldsymbol{\theta}_2)|| \leq 2\epsilon||\nabla_{\boldsymbol{\theta}}\boldsymbol{g}_1|| + M||\nabla_{\boldsymbol{\theta}}(\boldsymbol{g}_1 - \boldsymbol{g}_2)|| + L^b_{\boldsymbol{\theta}\boldsymbol{\theta}}||\boldsymbol{\theta}_1 - \boldsymbol{\theta}_2||$$
$$= 2\epsilon||\nabla_{\boldsymbol{\theta}}\boldsymbol{g}(\boldsymbol{x}; \boldsymbol{\theta}_1)|| + M||\nabla_{\boldsymbol{\theta}}\boldsymbol{g}(\boldsymbol{x}; \boldsymbol{\theta}_1) - \nabla_{\boldsymbol{\theta}}\boldsymbol{g}(\boldsymbol{x}; \boldsymbol{\theta}_2)|| + L^b_{\boldsymbol{\theta}\boldsymbol{\theta}}||\boldsymbol{\theta}_1 - \boldsymbol{\theta}_2||.$$

Note that $\boldsymbol{\theta}_1$ in the first term is arbitrarily chosen in (16). Therefore, this leads to the final inequality (15). □

**Theorem 1.** *Given input $\boldsymbol{x} \in \mathcal{X}$ and perturbation radius $\epsilon$, let $M$ be $\max_{\boldsymbol{x}' \in \mathbb{B}(\boldsymbol{x}, \epsilon)} ||\boldsymbol{x}'||$. For a linear relaxation-based method with the upper bound $\overline{s}_m(\boldsymbol{x}; \boldsymbol{\theta}) = \max_{\boldsymbol{x}' \in \mathbb{B}(\boldsymbol{x}, \epsilon)} \boldsymbol{g}^{(m)}(\boldsymbol{x}; \boldsymbol{\theta})^T \boldsymbol{x}' + b^{(m)}(\boldsymbol{x}; \boldsymbol{\theta})$, if $b^{(m)}$ satisfies Assumption 1 (1) for each $m$ and $\boldsymbol{p_s}$ satisfies Assumption 1 (2), then*

$$||\nabla_{\boldsymbol{\theta}}\mathcal{L}(\overline{\boldsymbol{s}}(\boldsymbol{x}; \boldsymbol{\theta}_1)) - \nabla_{\boldsymbol{\theta}}\mathcal{L}(\overline{\boldsymbol{s}}(\boldsymbol{x}; \boldsymbol{\theta}_2))||$$
$$\leq \max_m \left(2\epsilon||\nabla_{\boldsymbol{\theta}}\boldsymbol{g}^{(m)}(\boldsymbol{x}; \boldsymbol{\theta}_{1,2})|| + M||\nabla_{\boldsymbol{\theta}}\boldsymbol{g}^{(m)}(\boldsymbol{x}; \boldsymbol{\theta}_1) - \nabla_{\boldsymbol{\theta}}\boldsymbol{g}^{(m)}(\boldsymbol{x}; \boldsymbol{\theta}_2)|| + L^{(m)}||\boldsymbol{\theta}_1 - \boldsymbol{\theta}_2||\right)$$
$$(4)$$

*for any $\boldsymbol{\theta}_1, \boldsymbol{\theta}_2$, where $L^{(m)} = L^{b^{(m)}}_{\boldsymbol{\theta}\boldsymbol{\theta}} + L^{\boldsymbol{p_s}}_{\boldsymbol{\theta}}||\nabla_{\boldsymbol{\theta}}\overline{s}(\boldsymbol{x}; \boldsymbol{\theta}_{1,2})||$ and $\boldsymbol{\theta}_{1,2}$ can be any of $\boldsymbol{\theta}_1$ and $\boldsymbol{\theta}_2$.*

*Proof.* We simplify the notation $p_s$ as $p$. Then we have

$$||\nabla_{\boldsymbol{\theta}}\mathcal{L}(\overline{\boldsymbol{s}}(\boldsymbol{x};\boldsymbol{\theta}_1)) - \nabla_{\boldsymbol{\theta}}\mathcal{L}(\overline{\boldsymbol{s}}(\boldsymbol{x};\boldsymbol{\theta}_2))||$$
$$= ||\nabla_{\boldsymbol{\theta}}\overline{\boldsymbol{s}}(\boldsymbol{x};\boldsymbol{\theta}_1)\nabla_{\overline{\boldsymbol{s}}}\mathcal{L}(\overline{\boldsymbol{s}}(\boldsymbol{x};\boldsymbol{\theta}_1)) - \nabla_{\boldsymbol{\theta}}\overline{\boldsymbol{s}}(\boldsymbol{x};\boldsymbol{\theta}_2)\nabla_{\overline{\boldsymbol{s}}}\mathcal{L}(\overline{\boldsymbol{s}}(\boldsymbol{x};\boldsymbol{\theta}_2))||$$
$$= ||\sum_m \nabla_{\boldsymbol{\theta}}\overline{\boldsymbol{s}}_m(\boldsymbol{x};\boldsymbol{\theta}_1)(\boldsymbol{p}_m(\boldsymbol{x};\boldsymbol{\theta}_1) - \delta_{y,m}) - \nabla_{\boldsymbol{\theta}}\overline{\boldsymbol{s}}_m(\boldsymbol{x};\boldsymbol{\theta}_2)(\boldsymbol{p}_m(\boldsymbol{x};\boldsymbol{\theta}_2) - \delta_{y,m})||$$
$$= ||\nabla_{\boldsymbol{\theta}}\overline{\boldsymbol{s}}(\boldsymbol{x};\boldsymbol{\theta}_1)(\boldsymbol{p}(\boldsymbol{x};\boldsymbol{\theta}_1) - \boldsymbol{e}^{(y)}) - \nabla_{\boldsymbol{\theta}}\overline{\boldsymbol{s}}(\boldsymbol{x};\boldsymbol{\theta}_2)(\boldsymbol{p}(\boldsymbol{x};\boldsymbol{\theta}_2) - \boldsymbol{e}^{(y)})||$$
$$= ||\nabla_{\boldsymbol{\theta}}\overline{\boldsymbol{s}}(\boldsymbol{x};\boldsymbol{\theta}_1)\boldsymbol{p}(\boldsymbol{x};\boldsymbol{\theta}_1) - \nabla_{\boldsymbol{\theta}}\overline{\boldsymbol{s}}(\boldsymbol{x};\boldsymbol{\theta}_2)\boldsymbol{p}(\boldsymbol{x};\boldsymbol{\theta}_2)||$$
$$= ||\nabla_{\boldsymbol{\theta}}\overline{\boldsymbol{s}}(\boldsymbol{x};\boldsymbol{\theta}_1)\boldsymbol{p}(\boldsymbol{x};\boldsymbol{\theta}_1) - \nabla_{\boldsymbol{\theta}}\overline{\boldsymbol{s}}(\boldsymbol{x};\boldsymbol{\theta}_1)\boldsymbol{p}(\boldsymbol{x};\boldsymbol{\theta}_2) + \nabla_{\boldsymbol{\theta}}\overline{\boldsymbol{s}}(\boldsymbol{x};\boldsymbol{\theta}_1)\boldsymbol{p}(\boldsymbol{x};\boldsymbol{\theta}_2) - \nabla_{\boldsymbol{\theta}}\overline{\boldsymbol{s}}(\boldsymbol{x};\boldsymbol{\theta}_2)\boldsymbol{p}(\boldsymbol{x};\boldsymbol{\theta}_2)||$$
$$= ||\nabla_{\boldsymbol{\theta}}\overline{\boldsymbol{s}}(\boldsymbol{x};\boldsymbol{\theta}_1)(\boldsymbol{p}(\boldsymbol{x};\boldsymbol{\theta}_1) - \boldsymbol{p}(\boldsymbol{x};\boldsymbol{\theta}_2)) + (\nabla_{\boldsymbol{\theta}}\overline{\boldsymbol{s}}(\boldsymbol{x};\boldsymbol{\theta}_1) - \nabla_{\boldsymbol{\theta}}\overline{\boldsymbol{s}}(\boldsymbol{x};\boldsymbol{\theta}_2))\boldsymbol{p}(\boldsymbol{x};\boldsymbol{\theta}_2)||$$
$$\leq ||\nabla_{\boldsymbol{\theta}}\overline{\boldsymbol{s}}(\boldsymbol{x};\boldsymbol{\theta}_1)||\,||\boldsymbol{p}(\boldsymbol{x};\boldsymbol{\theta}_1) - \boldsymbol{p}(\boldsymbol{x};\boldsymbol{\theta}_2)|| + \max_m ||\nabla_{\boldsymbol{\theta}}\overline{\boldsymbol{s}}_m(\boldsymbol{x};\boldsymbol{\theta}_1) - \nabla_{\boldsymbol{\theta}}\overline{\boldsymbol{s}}_m(\boldsymbol{x};\boldsymbol{\theta}_2)||$$
$$\leq ||\nabla_{\boldsymbol{\theta}}\overline{\boldsymbol{s}}(\boldsymbol{x};\boldsymbol{\theta}_1)||L_{\boldsymbol{\theta}}^p||\boldsymbol{\theta}_1 - \boldsymbol{\theta}_2|| + \max_m ||\nabla_{\boldsymbol{\theta}}\overline{\boldsymbol{s}}_m(\boldsymbol{x};\boldsymbol{\theta}_1) - \nabla_{\boldsymbol{\theta}}\overline{\boldsymbol{s}}_m(\boldsymbol{x};\boldsymbol{\theta}_2)||$$
$$\leq \max_m \left( 2\epsilon||\nabla_{\boldsymbol{\theta}}\boldsymbol{g}^{(m)}(\boldsymbol{x};\boldsymbol{\theta}_{1,2})|| + M||\nabla_{\boldsymbol{\theta}}\boldsymbol{g}^{(m)}(\boldsymbol{x};\boldsymbol{\theta}_1) - \nabla_{\boldsymbol{\theta}}\boldsymbol{g}^{(m)}(\boldsymbol{x};\boldsymbol{\theta}_2)|| + L^{(m)}||\boldsymbol{\theta}_1 - \boldsymbol{\theta}_2|| \right)$$

$\square$

## G   LEARNING CURVE FOR $\epsilon_{train}$

Figure 9 shows the learning curves for the target perturbation $\epsilon_{train}$ during the ramp-up period, while Figure 1 shows the corresponding curves for the scheduled value of $\epsilon$. The two figures use the same settings in Appendix A.1.

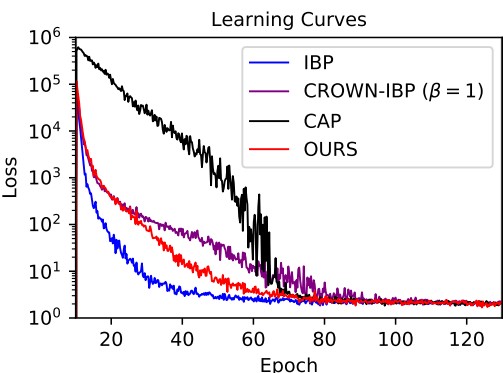

Figure 9: The learning curves for the target perturbation $\epsilon_{train}$ during the ramp-up period.

# H  SMOOTHNESS

We empirically measure the non-smoothness of the loss landscape with the difference between the two consecutive loss gradients at $\boldsymbol{\theta}_1 = \boldsymbol{\theta}(0)$ and $\boldsymbol{\theta}_2 = \boldsymbol{\theta}(1)$ in (8), says gradient difference ($\equiv ||\nabla_\theta \mathcal{L}(\boldsymbol{x}; \boldsymbol{\theta}(0)) - \nabla_\theta \mathcal{L}(\boldsymbol{x}; \boldsymbol{\theta}(1))||$). It is highly related to the ratio of the number of unstable ReLUs (nonlinearity of the classifier) as shown in Figure 10.

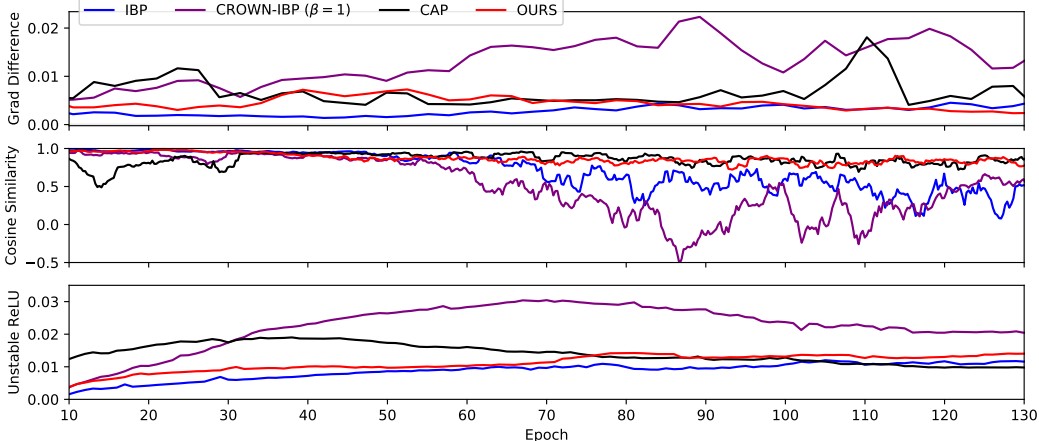

Figure 10: (*Top*) Gradient difference and (*Middle*) cosine similarities between two consecutive loss gradients, and (*Bottom*) the ratio of the number of unstable ReLUs during the ramp-up period.

## I   MODE CONNECTIVITY

In this section, we check the mode connectivity (Garipov et al., 2018) between two models that are trained using certifiable training methods. Mode connectivity is a framework that investigates the connectedness between two models by finding a high accuracy curve between those models. It enables us to understand the loss surface of neural networks.

Let $\boldsymbol{w}_0$ and $\boldsymbol{w}_1$ be two sets of weight corresponding to two different well-trained neural networks. Moreover, let $\phi_{\boldsymbol{\theta}_c}(t)$ with $t \in [0, 1]$ be a continuous piece-wise smooth parametric curve with parameters $\boldsymbol{\theta}_c$ such that $\phi_{\boldsymbol{\theta}_c}(0) = \boldsymbol{w}_0$ and $\phi_{\boldsymbol{\theta}_c}(1) = \boldsymbol{w}_1$. To find a low-loss path between $\boldsymbol{w}_0$ and $\boldsymbol{w}_1$, Garipov et al. (2018) suggested to find the parameter $\boldsymbol{\theta}_c$ that minimizes the expectation of a loss $\ell(\boldsymbol{w})$ over a distribution $q_{\boldsymbol{\theta}_c}(t)$ on the curve,

$$L(\boldsymbol{\theta}_c) = \mathbb{E}_{t \sim q_{\boldsymbol{\theta}_c}(t)}[\ell(\phi_{\boldsymbol{\theta}_c}(t)].$$

To optimize $L(\boldsymbol{\theta}_c)$ for $\boldsymbol{\theta}_c$, we use uniform distribution $U[0, 1]$ as $q_{\boldsymbol{\theta}_c}(t)$ and Bezier curve (Farouki, 2012) as $\phi_{\boldsymbol{\theta}_c}(t)$, which provides a convenient parameterization of smoothness on the paths connecting two end points ($\boldsymbol{w}_0$ and $\boldsymbol{w}_1$) as follows:

$$\phi_{\boldsymbol{\theta}_c}(t) = (1 - t)^2 \boldsymbol{w}_0 + 2t(1 - t)\boldsymbol{\theta}_c + t^2 \boldsymbol{w}_1, 0 \le t \le 1.$$

A path $\phi_{\boldsymbol{\theta}_c}$ is said to have a barrier if $\exists t$ such that $\ell(\phi_{\boldsymbol{\theta}_c}(t)) > \max\{\ell(\boldsymbol{w}_0), \ell(\boldsymbol{w}_1)\}$. The existence of a barrier suggests the modes of two well-trained models are not connected by the path in terms of the given loss function $\ell$ (Zhao et al., 2020).

We test the mode connectivity between the models trained with IBP, CROWN-IBP, and OURS. For example, to check the mode connectivity between two different models trained with CROWN-IBP and IBP, we use the loss function used on each model as a user-specified loss for training the parametric curve $\phi_{\boldsymbol{\theta}_c}$. Therefore, we can obtain two curves as depicted in Figure 11, 12, and 13 for each pair of models. Here, we use the identical settings in Appendix A.1.

Figure 11 shows the mode-connectivity between CROWN-IBP and IBP. We use CROWN-IBP loss as user-specific loss in Figure 11a and IBP loss in Figure 11b. In this figure, we find that using CROWN-IBP loss (11a), there exists a barrier between the two models. This suggests they are not connected by the path in terms of CROWN-IBP loss. However, with IBP loss, there is no loss barrier separating the two models. This indicates that using CROWN-IBP, it is hard to optimize the parameters from $\boldsymbol{w}_0$ to $\boldsymbol{w}_1$, but IBP can.

Figure 12 shows the mode-connectivity results on IBP and OURS. We find that two models are not connected to each other using either IBP bound or OURS bound, since there exists a barrier in both curves. In this figure, we can also notify that OURS has tighter bounds than IBP because the value of the loss function using OURS is lower than that of IBP.

Finally, Figure 13 illustrates the mode connectivity between CROWN-IBP and OURS. Using CROWN-IBP as a user-specified loss function, we can find that the robust loss on the curve is higher than that of the end points. However, when OURS is used as a loss function, the robust loss generally decreases as the $t$ increases. It shows that OURS has much favorable loss landscape compared to CROWN-IBP. In addition, we can find that OURS has a tighter bound than CROWN-IBP, since the value of the robust loss using OURS is lower than CROWN-IBP.

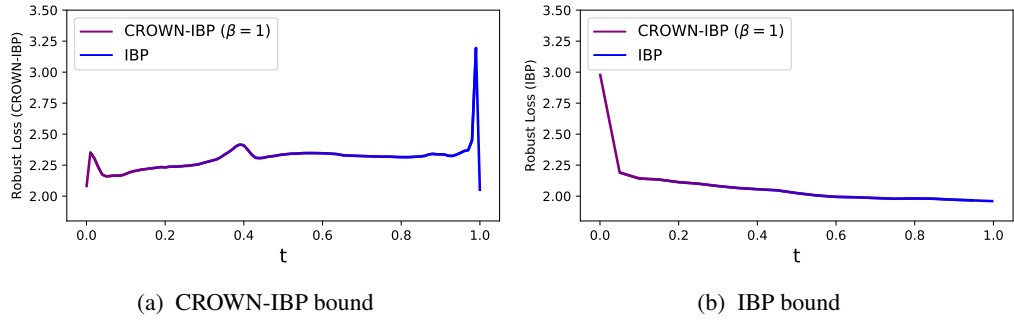

(a) CROWN-IBP bound

(b) IBP bound

Figure 11: Mode connectivity between CROWN-IBP and IBP, where $w_0$ and $w_1$ are well-trained models using CROWN-IBP bound and IBP bound, respectively. $\theta_c$ is trained using CROWN-IBP (11a) and IBP (11b), respectively.

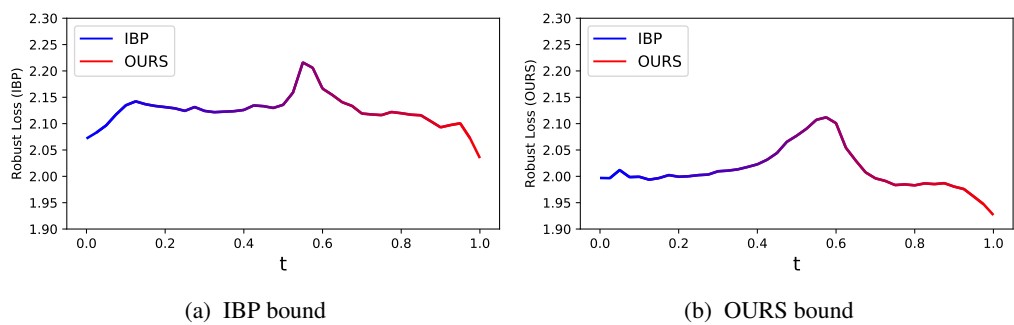

(a) IBP bound

(b) OURS bound

Figure 12: Mode connectivity between IBP and OURS, where $w_0$ and $w_1$ are well-trained models using IBP bound and OURS bound, respectively. $\theta_c$ is trained using IBP (12a) and OURS (12b), respectively.

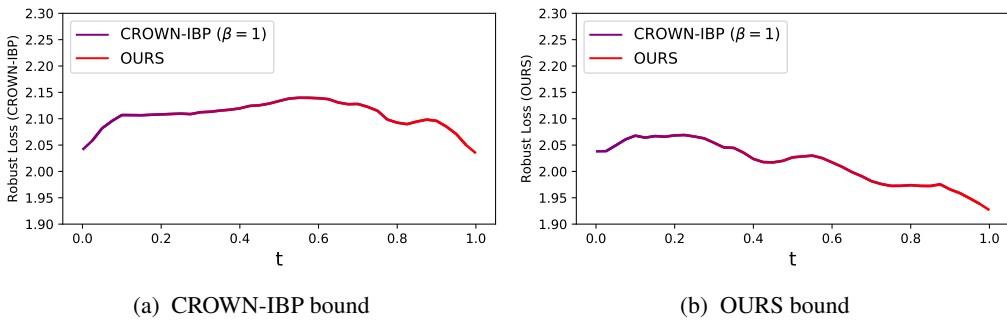

(a) CROWN-IBP bound

(b) OURS bound

Figure 13: Mode connectivity between CROWN-IBP and OURS, where $w_0$ and $w_1$ are well-trained models using CROWN-IBP bound and OURS bound, respectively. $\theta_c$ is trained using CROWN-IBP (13a) and OURS (13b), respectively.

## J  RELU

In this section, we investigate how pre-activation bounds $u$ and $l$ for the activation layer change during training. For each activation node, it is said to be "active" when the pre-activation bounds are both positive ($0 < l \le u$), "unstable" when they span zero ($l \le 0 \le u$), and "dead" when they are both negative ($l \le u < 0$).

Figure 14 shows the ratios of the number of active and dead ReLUs during the ramp-up period. Notably, we find that CROWN-IBP has more active ReLUs during training compared to the other three methods. Simultaneously, CROWN-IBP has the lowest ratio of dead ReLUs.

Figure 15 shows the numbers of active, unstable, and dead ReLUs during the ramp-up period. We find that in CROWN-IBP, the number of unstable and active ReLUs increases as the number of dead ReLUs decreases. This indicates that a number of dead ReLUs change to unstable ReLUs as the training $\epsilon$ increases. However, in the other methods, the number of unstable ReLUs is consistently small, while the number of active ReLUs decreases as the number of dead ReLUs increases.

Figure 16 depicts the histograms of the distribution of the slope $\frac{u^+}{u^+ - l^-}$ of the unstable ReLUs during the ramp-up period. In the early stages of CAP training, the slope distribution is concentrated around 0.4. However as the training progresses with a larger $\epsilon$, the histogram distribution moves to left, which indicates unstable ReLUs change to dead ReLUs. It is consistent with the results in Figure 15c. On the other hand, in the case of CROWN-IBP, the histogram distribution moves to right during training. It is the same with the results in Figure 15b, which shows that number of active ReLUs increases during training.

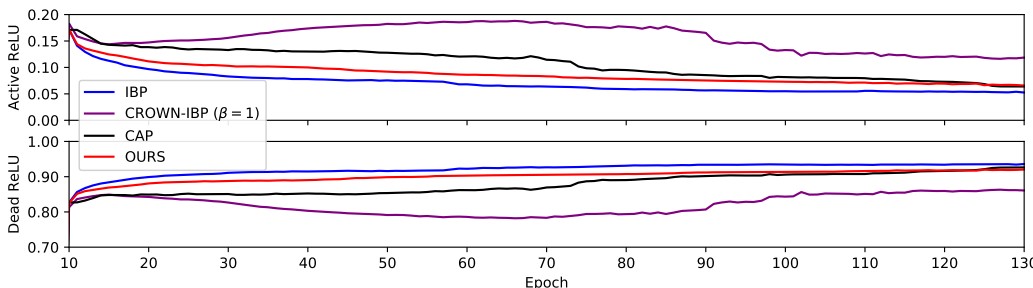

Figure 14:  The ratio of the number of active (*top*) and dead (*bottom*) ReLUs during the ramp-up period.

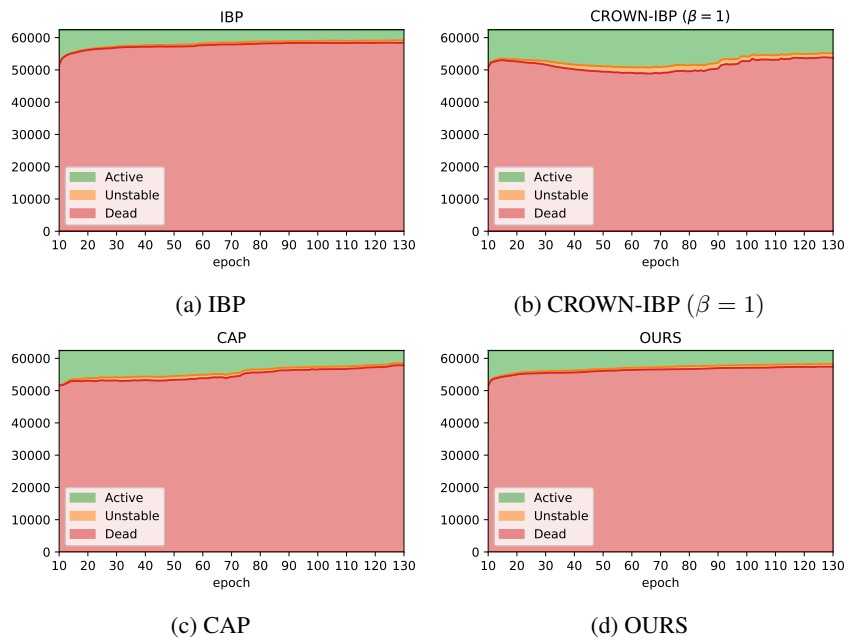

Figure 15: Number of active (Green), unstable (Orange), and dead (Red) ReLUs.

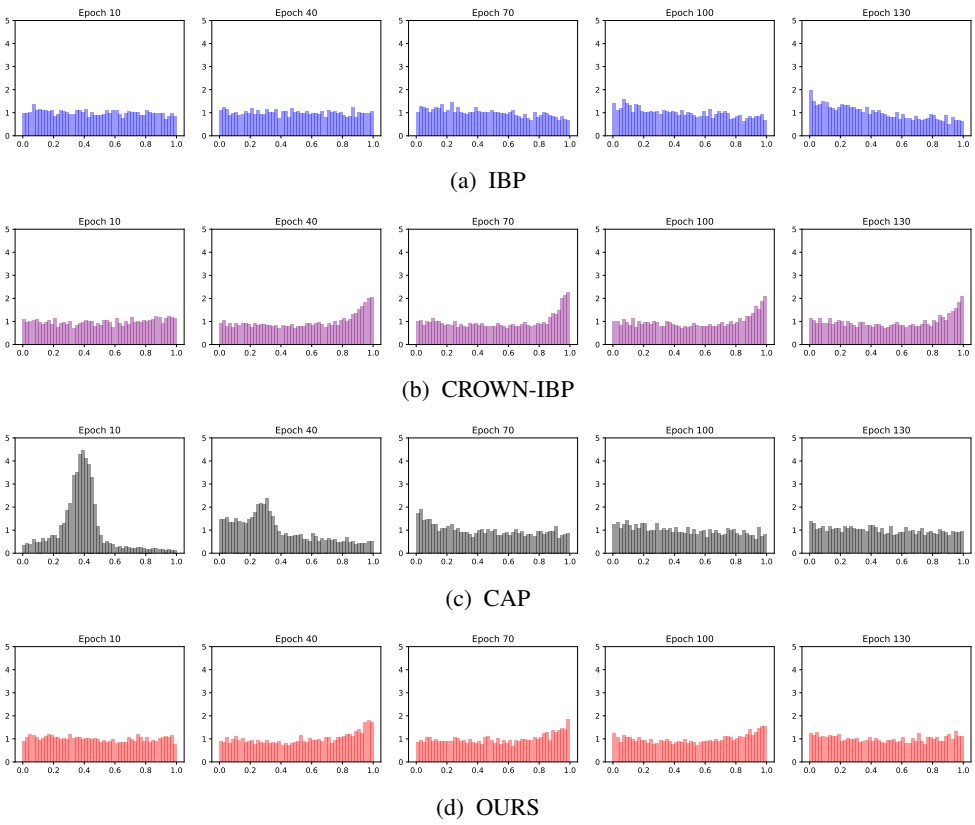

Figure 16: Histograms of the distribution of the slope $\frac{u^+}{u^+ - l^-}$ when $l \leq 0 \leq u$ during the ramp-up period.

## K  COMPARISON WITH OTHER PRIOR WORK

All experiments and results (except for Table 4) in this paper are based on our own reimplementation. For the unimplemented prior work, we compare to the best reported results in the literature in Table 4. We note that the results in Xiao et al. (2018) and Balunovic & Vechev (2019) are evaluated with a MILP based exact verifier (Tjeng et al., 2017).

Table 4: Test errors (Standard / Verified error) compared to the best errors reported in the literature. Bold numbers are the lowest verified error.

| Data | $\epsilon(l_\infty)$ | Xiao et al. (2018) | Mirman et al. (2018) | Balunovic & Vechev (2019) | OURS |
|---|---|---|---|---|---|
| MNIST | 0.1 | 1.32 / 4.87 | 1.3 / 4.2 | 0.8 / 2.9 | 1.09 / **2.28** |
| | 0.3 | 2.67 / 19.32 | 3.4 / 10.7 | 2.7 / 14.3 | 2.42 / **7.84** |
| CIFAR-10 | $2/255$ | 38.88 / 54.07 | 37.7 / 54.5 | 21.6 / **39.5** | 31.49 / 49.42 |
| | $8/255$ | 59.55 / 79.72 | 53.8 / 72.8 | 48.3 / 72.5 | 56.01 / **69.70** |

## L  $\beta$- AND $\kappa$-SCHEDULINGS

Table 5 shows the evaluation results of the models as in Table 1 but trained with different $\kappa$-scheduling (from 0 to 0). Table 6 shows the evaluation results of the proposed models trained with different $\kappa$- and $\beta$-schedulings.

Table 5: Test errors (Standard / PGD / Verified error) of IBP, CROWN-IBP ($\beta = 1$), CAP, and OURS on MNIST, CIFAR-10, and SVHN. See Appendix A for all the other settings, same as in Table 1. Bold and underline numbers are the first and second lowest verified error.

| Data | $\epsilon(l_\infty)$ | IBP | CROWN-IBP ($\beta = 1$) | CAP | OURS |
|---|---|---|---|---|---|
| **MNIST** | $\epsilon = 0.1$ | 1.25 / 2.31 / 3.10 | 1.23 / 2.19 / 2.75 | 0.80 / 1.73 / 3.19 | 1.09 / 1.86 / **2.28** |
| | $\epsilon = 0.2$ | 1.95 / 2.95 / 6.28 | 2.89 / 5.32 / 7.61 | 3.22 / 6.72 / 11.06 | 1.70 / 3.37 / **4.78** |
| | $\epsilon = 0.3$ | 3.67 / 5.55 / 9.74 | 6.11 / 11.33 / 17.51 | 19.19 / 35.84 / 47.85 | 3.39 / 4.85 / **9.12** |
| | $\epsilon = 0.4$ | 3.67 / 6.55 / 16.55 | 6.11 / 15.34 / 26.72 | - | 3.39 / 5.88 / **15.04** |
| **CIFAR 10** | $\epsilon = 2/255$ | 43.60 / 52.62 / 56.58 | 32.15 / 42.67 / 49.36 | 28.80 / 38.95 / **48.50** | 32.04 / 43.13 / 49.62 |
| | $\epsilon = 4/255$ | 53.89 / 62.58 / 65.14 | 45.05 / 56.46 / 63.04 | 40.78 / 52.62 / 61.88 | 43.15 / 54.85 / **61.31** |
| | $\epsilon = 6/255$ | 61.37 / 68.64 / 70.82 | 53.87 / 65.03 / 71.08 | 49.20 / 60.85 / 69.03 | 50.99 / 62.23 / **67.59** |
| | $\epsilon = 8/255$ | 64.11 / 70.68 / 72.99 | 60.96 / 70.52 / 75.68 | 56.77 / 66.78 / 73.02 | 56.35 / 67.06 / **70.56** |
| | $\epsilon = 16/255$ | 69.74 / 76.66 / 79.86 | 79.14 / 83.64 / 84.36 | 75.11 / 80.67 / 82.07 | 66.96 / 75.63 / **78.08** |
| **SVHN** | $\epsilon = 0.01$ | 20.19 / 34.57 / 44.25 | 16.66 / 30.05 / 38.15 | 16.88 / 30.16 / **37.09** | 15.46 / 29.34 / 38.57 |

## M  ONE-STEP VS MULTI-STEP

To get a tighter bound, we propose multi-step version of (6) as follows:

$$\underline{a}_{t+1} = \Pi_{[0,1]^n} \left( \underline{a}_t - \alpha \text{sign}(\nabla_{\underline{a}} \mathcal{L}(\overline{s}(x, y, \epsilon; \theta, \phi), y)) \right). \tag{17}$$

We compare the original 1-step method ($\alpha \geq 1$) to 7-step ($t = 7$) method with $\alpha = 0.1$. The results are summarized in Table 7. We found no significant difference between two methods even though multi-step takes multiple times with multi-step. Therefore, we decide to focus on one-step method.

Table 6: Test errors of OURS with different $\beta$- and $\kappa$-scheduling on MNIST and CIFAR-10.

| Data | $\epsilon(l_\infty)$ | $\mathbf{OURS}_{1 \to 1}$ ($\kappa = 1 \to 0$) | | | $\mathbf{OURS}_{1 \to 0}$ ($\kappa = 1 \to 0$) | | | $\mathbf{OURS}_{1 \to 1}$ ($\kappa = 0 \to 0$) | | | $\mathbf{OURS}_{1 \to 0}$ ($\kappa = 0 \to 0$) | | |
|---|---|---|---|---|---|---|---|---|---|---|---|---|---|
| | | Standard | PGD | Verfied | Standard | PGD | Verfied | Standard | PGD | Verfied | Standard | PGD | Verfied |
| **MNIST** | $\epsilon = 0.1$ | 1.09 | 1.77 | 2.36 | 1.29 | 2.29 | 3.58 | 1.09 | 1.86 | 2.28 | 1.15 | 2.03 | 3.53 |
| | $\epsilon = 0.2$ | 1.70 | 3.44 | 4.34 | 1.61 | 3.09 | 5.71 | 1.70 | 3.37 | 4.78 | 1.64 | 2.57 | 5.43 |
| | $\epsilon = 0.3$ | 3.49 | 5.59 | 9.79 | 2.42 | 4.37 | 7.84 | 3.39 | 4.85 | 9.12 | 2.44 | 4.41 | 8.00 |
| | $\epsilon = 0.4$ | 3.49 | 6.77 | 15.42 | 2.42 | 5.68 | 13.72 | 3.39 | 5.88 | 15.04 | 2.44 | 5.29 | 13.84 |
| **CIFAR 10** | $\epsilon = 2/255$ | 31.49 | 42.73 | 49.42 | 37.77 | 48.30 | 54.43 | 32.04 | 43.13 | 49.62 | 38.58 | 48.59 | 54.63 |
| | $\epsilon = 8/255$ | 56.01 | 66.17 | 69.70 | 58.87 | 67.76 | 71.50 | 56.35 | 67.06 | 70.56 | 58.90 | 67.81 | 70.99 |
| | $\epsilon = 16/255$ | 65.39 | 75.39 | 77.87 | 66.24 | 74.69 | 78.66 | 66.96 | 75.63 | 78.08 | 66.76 | 75.17 | 77.99 |

Table 7: Test errors of OURS with different numbers of gradient update steps in (17) on CIFAR-10. Here, we use $\kappa$-scheduling from 0 to 0.

| Data | $\epsilon(l_\infty)$ | **OURS (1-step)** | | | **OURS (7-step)** | | |
|---|---|---|---|---|---|---|---|
| | | Standard | PGD | Verfied | Standard | PGD | Verfied |
| **CIFAR-10** | $\epsilon = 2/255$ | 32.04 | 43.11 | 49.62 | 31.40 | 42.30 | 49.20 |
| | $\epsilon = 8/255$ | 56.35 | 67.03 | 70.56 | 54.44 | 66.29 | 71.53 |

# N  TRAIN WITH $\epsilon_{train} \geq \epsilon_{test}$

## N.1  $\epsilon_{train} \geq \epsilon_{test}$ ON MNIST

Zhang et al. (2019b) and Gowal et al. (2018) observed that IBP performs better when using $\epsilon_{train} \geq \epsilon_{test}$ than $\epsilon_{train} = \epsilon_{test}$. Figure 8 shows the results with different $\epsilon_{train}$'s for each $\epsilon_{test}$. The overfitting issue is more prominent in the case of IBP and CROWN-IBP$_{1\to0}$ than the proposed method and CROWN-IBP$_{1\to1}$. However, using larger perturbations compromises the standard accuracy, and thus it is desirable to use smaller $\epsilon_{train}$.

Table 8: Comparison of the performance (Standard / PGD / Verified error) depending on various $\epsilon_{train}$. Here, we use $\kappa$-scheduling from 0 to 0.

| Data | $\epsilon_{test}$ | $\epsilon_{train}$ | IBP | CROWN-IBP$_{1\to1}$ | OURS | CROWN-IBP$_{1\to0}$ |
|---|---|---|---|---|---|---|
| **MNIST** | 0.2 | 0.2 | 1.25 / 3.39 / 7.77 | 1.23 / 3.48 / 7.64 | 1.09 / 3.17 / 6.29 | 1.13 / 2.85 / 5.89 |
| | | 0.3 | 1.95 / 2.93 / 6.28 | 2.89 / 5.32 / 7.61 | 1.70 / 3.37 / 4.76 | 1.48 / 2.73 / 4.79 |
| | | 0.4 | 3.67 / 4.77 / 6.36 | 6.11 / 9.08 / 12.71 | 3.49 / 4.72 / 6.36 | 2.37 / 3.26 / 4.64 |
| | 0.3 | 0.3 | 1.95 / 3.31 / 12.90 | 2.89 / 7.35 / 14.97 | 1.70 / 4.82 / 9.20 | 1.48 / 3.52 / 9.40 |
| | | 0.4 | 3.67 / 5.55 / 9.74 | 6.11 / 11.33 / 17.51 | 3.49 / 5.59 / 9.79 | 2.37 / 3.63 / 7.22 |

## N.2  $\epsilon_{train} = 1.1\epsilon_{test}$ ON CIFAR-10

As mentioned in Gowal et al. (2018), we also train with $\epsilon_{train} = 1.1\epsilon_{test}$ on CIFAR-10. The results are shown in Table 9. They attain slightly improved performances in $2/255$, but not in $8/255$ and larger $\epsilon$.

Table 9: Comparison of the performance (Standard / PGD / Verified error) of the models trained with $\epsilon_{train}$ and $1.1\epsilon_{train}$. Here, we use $\kappa$-scheduling from 0 to 0.

| Data | $\epsilon_{test}$ | $\epsilon_{train}$ | IBP | CROWN-IBP$_{1\to1}$ | OURS | CROWN-IBP$_{1\to0}$ |
|---|---|---|---|---|---|---|
| **CIFAR 10** | $2/255$ | $2/255$ | 43.6 / 52.71 / 56.58 | 32.15 / 42.67 / 49.36 | 32.04 / 43.13 / 49.62 | 37.25 / 47.19 / 52.53 |
| | | $2.2/255$ | 44.78 / 52.62 / 55.78 | 33.23 / 43.11 / 49.18 | 33.04 / 43.70 / 48.60 | 38.42 / 47.80 / 52.53 |
| | $8/255$ | $8/255$ | 64.11 / 70.68 / 72.99 | 60.96 / 70.52 / 75.68 | 56.35 / 67.06 / 70.56 | 56.95 / 67.89 / 70.43 |
| | | $8.8/255$ | 64.54 / 70.30 / 72.40 | 61.48 / 70.58 / 75.17 | 58.28 / 67.50 / 70.52 | 59.37 / 68.51 / 70.71 |

# O  TRAINING TIME

All the training times are measured on a single TITAN X (Pascal) on Medium for CIFAR-10. We train with a batch size of 128 for OURS, CROWN-IBP$_{1\to1}$ and IBP, but with a batch size of 32 for CAP due to its high memory cost. For CAP, we use random projection of 50 dimensions.

- OURS: 115.9 sec / epoch
- CROWN-IBP$_{1\to1}$: 51.68 sec / epoch
- IBP: 14.85 sec / epoch
- CAP (batch size 32, 1 GPU): 751.0 sec / epoch
- CAP (batch size 64, 1 GPU): 724.6 sec / epoch
- CAP (batch size 128, 2 GPUs): 387.9 sec / epoch

# P  LOSS AND TIGHTNESS VIOLIN PLOTS

We plot the equivalent tightness violin plots in Section 6 for models trained with other methods. The proposed method achieves the best results in terms of loss and tightness followed by CROWN-IBP, CAP-IBP, and RANDOM. Figure 17 (a)-(b), (c)-(d), and (e)-(f) show the tightness evaluated on the model trained by CROWN-IBP$_{1\rightarrow 0}$, CROWN-IBP$_{1\rightarrow 1}$ and IBP, respectively.

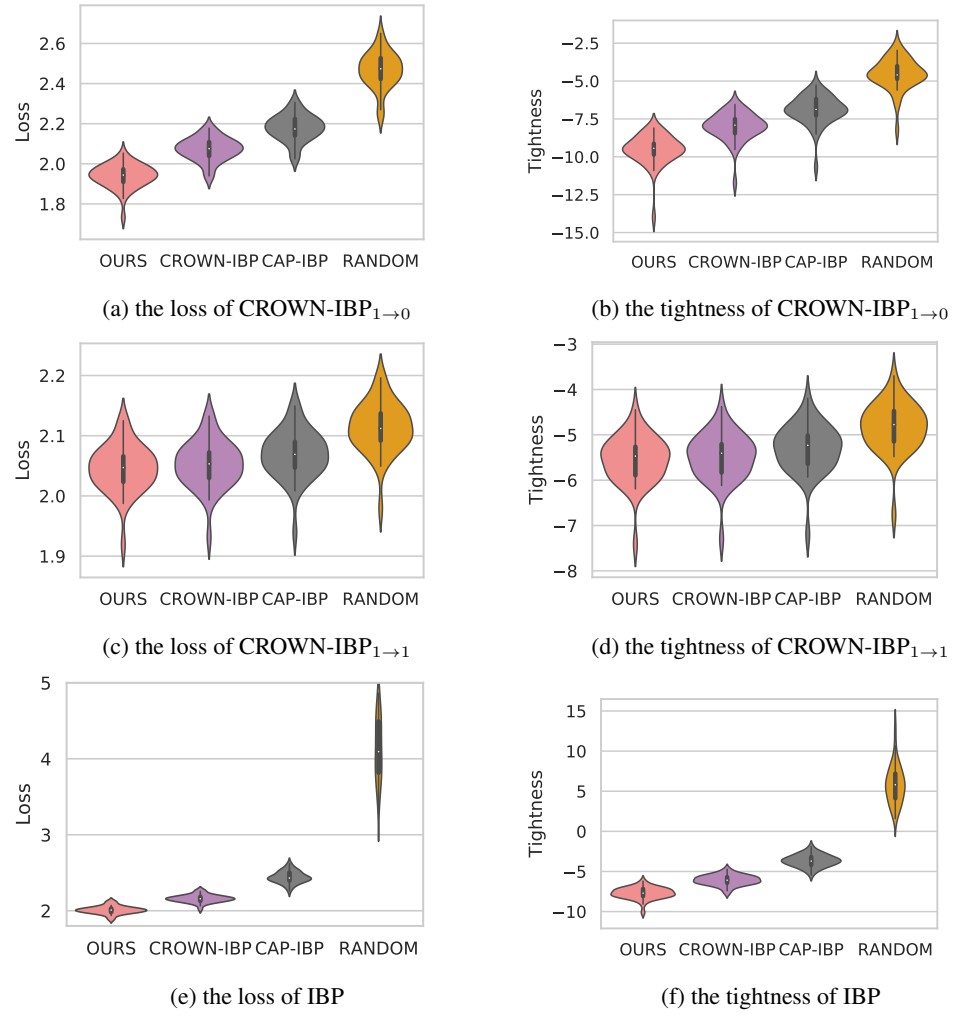

Figure 17: Violin plots of the test loss (*Left Column*) and of tightness (*Right Column*) for various linear relaxations same as in Section 6. Lower is better.

## Q COMPARISON WITH CAP-IBP

As in section E, we train a model with CAP-IBP and compare with the proposed method and CROWN-IBP ($\beta = 1$). Figure 18 shows that CAP-IBP has gradient differences (defined in Section H) larger than the proposed method and smaller than CROWN-IBP ($\beta = 1$), which leads to a performance between the proposed method and CROWN-IBP ($\beta = 1$) (see Table 3). CAP-IBP has looser bounds than CROWN-IBP ($\beta = 1$) as shown in Figure 4 and Figure 17, but with a relatively more smooth landscape, it can achieve a better performance than CROWN-IBP ($\beta = 1$).

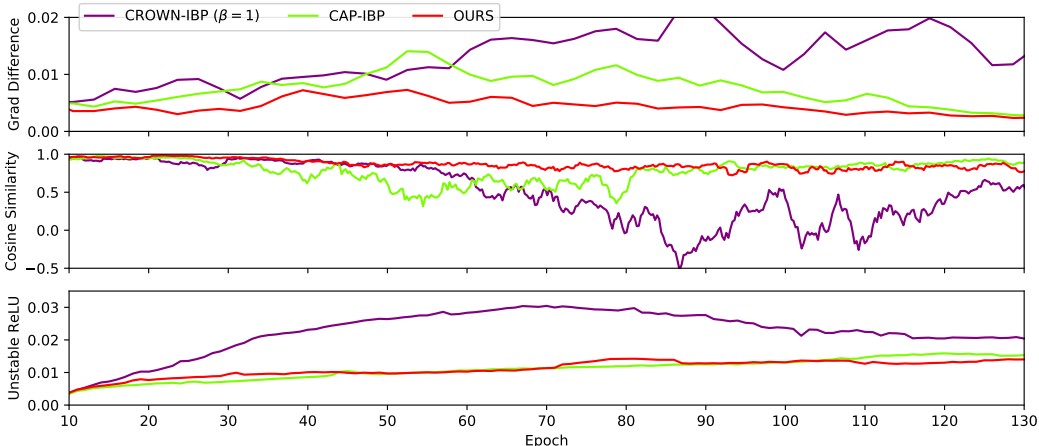

Figure 18: (*Top*) Gradient difference and (*Middle*) cosine similarities between two consecutive loss gradients, and (*Bottom*) the ratio of the number of unstable ReLUs during the ramp-up period.

# R  RELU STABILITY

To see the effect of unstable ReLUs on smoothness, we adopt the ReLU stability loss (RS loss) $\mathcal{L}_{RS}(\boldsymbol{u}, \boldsymbol{l}) = -tanh(1 + \boldsymbol{u} \cdot \boldsymbol{l})$ as a regularizer (Xiao et al., 2018). We use $\mathcal{L} + \lambda\mathcal{L}_{RS}$ as a loss and run CROWN-IBP ($\beta = 1$) with various $\lambda$ settings. We plot the smoothness and the tightness in Figure 19 and Figure 20 on $\lambda = 0$, $\lambda = 0.01$, $\lambda = 10$.

We found that small $\lambda$ suggested in Xiao et al. (2018) has no effect on reducing the number of unstable ReLUs since certifiable methods have smaller unstable ReLUs as shown in Figure 15, and thus not on improving the smoothness. By increasing $\lambda$, we observed that RS successfully reduces the number of unstable ReLUs with $\lambda = 10$. Figure 19 shows that large $\lambda$ leads to a better loss variation and gradient difference. This supports that unstable ReLUs are closely related to the smoothness of the loss landscape. However, as Xiao et al. (2018) mentioned "placing too much weight on RS Loss can decrease the model capacity, potentially lowering the provable adversarial accuracy", the models trained with a large $\lambda \geq 1$ couldn't obtain a tightness of the upper bound and significant improvement on robustness as illustrated in Figure 20. The test errors (Standard / PGD / Verified) are 0.6278 / 0.7189 / 0.7634 on $\lambda = 0.01$ and 0.6090 / 0.7085 / 0.7600 on $\lambda = 10$.

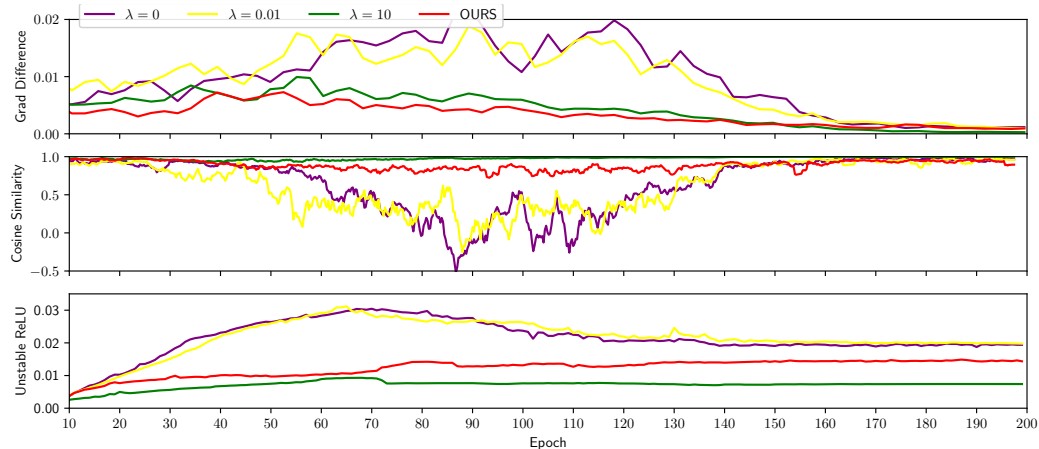

Figure 19: (*Top*) Gradient difference and (*Middle*) cosine similarities between two consecutive loss-gradients, and (*Bottom*) the ratio of the number of unstable ReLUs on CROWN-IBP ($\beta = 1$) with $\lambda = 0$, $\lambda = 0.01$, $\lambda = 10$ and OURS.

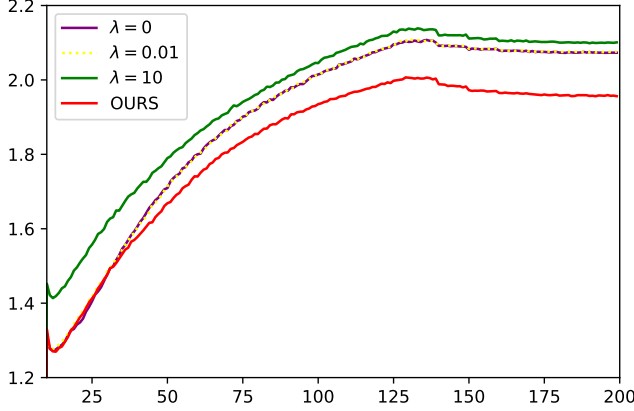

Figure 20: Robust loss of CROWN-IBP ($\beta = 1$) with $\lambda = 0$, $\lambda = 0.01$, $\lambda = 10$ and OURS during training.

