# OpenReview forum: "Loss Landscape Matters: Training Certifiably Robust Models with Favorable Loss Landscape"
_ICLR.cc/2021/Conference — Reject_

### Official Review · AnonReviewer1 · 2020-10-28
**Important problem and interesting method; but discussions and experiments are limited**

**Rating:** 4
**Confidence:** 5

**Review:**

This paper studies why training with looser bounds (IBP) can outperform tighter linear relaxation based methods in certified defense. The authors argue that this is because IBP has a smoother loss landscape compared to linear relaxation based methods. Then the paper proposes to optimize the lower bound in the CROWN relaxation for unstable ReLU neurons during training, for tighter bounds and a smoother loss landscape.

Pros:
* This paper studies an important problem in certified defense and tries to improve linear relaxation based certified defense methods, especially CROWN.
* This paper novelly proposes to optimize the lower bound in the relaxation for unstable ReLU neurons to train tighter bounds in certified defense.
* The paper showed some improvement on test verifier error, compared to CROWN-IBP(1->1), and it also showed improvement on tightness and loss smoothness.

Cons:
* The link between the motivation and the proposed method is not well justified. Paragraph “Favorable Landscape” somewhat explains the connection between optimizing the lower bound in relaxation and improving loss landscape. But it states that the benefit comes from preferring dead ReLU neurons to unstable ones. If the proposed method is actually trying to make more unstable neurons dead, the paper does not discuss or compare with paper Xiao et al., 2018 (https://arxiv.org/pdf/1809.03008.pdf) which directly adds a regularization to induce ReLU stability.
* It does not seem to make sense that the proposed method can prefer to have dead neurons via optimizing the relaxation for unstable neurons. With such an optimization, bounds of unstable neurons become tighter, so how can such tighter bounds make the model favor unstable neurons less than CROWN-IBP with relatively looser bounds?
* As the paper mentions that CROWN-IBP does not penalize unstable ReLU neurons with (|u|<=|l|), but how about just make the lower bound equal to the upper bound as Fast-lin (Weng et al., 2018, https://arxiv.org/abs/1804.09699) does, so the lower bound is non-zero for unstable neurons?
* An improvement over CROWN-IBP(1->1) is shown in the paper. However, the proposed method does not make enough improvement over CROWN-IBP(1->0). The improvement on CIFAR10 with eps=8/255 or 16/255 is very small (error 69.70 v.s. 69.97, 77.87 v.s. 77.88), while there is a large performance drop on MNIST with eps=0.3/0.4 (increased error 7.07->9.79 and 12.35->15.42). What will the performance of the proposed method look like if a changing $\beta$ is used in training?
* This paper missed some previous works, e.g., Xiao et al., 2018 and Weng et al., 2018 as mentioned above. More discussions or comparisons with them may be needed.

==========================================================================================

Update after rebuttal:

Thanks for the detailed author response. I think the paper is interesting in providing an analysis on how optimizing the linear relaxation in CROWN (although the verification method itself seems to be similar as the one in Fastened CROWN in AAAI 2020) can lead to better loss smoothness and tightness, which seems to improve the performance of certified training. The author replies have addressed some of my concerns in my initial review. However, there are still some outstanding concerns:

1. After more consideration, I think the “More favorable landscape” paragraph is still insufficient to address the second point in my above cons. The author response argues that some “$(p/q)$” have looser or tighter bounds, but these are considered for relaxation *locally*, not the tightness on the final output, while the relaxation optimized in the paper is to tighten the final output. Thus it remains unclear why tighter final bounds with improvement from the unstable ReLU neurons makes the model favor unstable neurons less.

2. AnonReviewer2 has reminded me that the “Fastened CROWN” work had a similar method about optimizing the lower bound of the linear relaxation in verification, which seems to be very similar to the method proposed in this paper, in terms of the verification part in certified training. Although this paper focuses on certified training and has some different analysis, the major modification on the method is still on the verification part, and thus I agree that a discussion on the comparison with Fastened CROWN should not be missed. The authors did not add it in the discussion period.

3. It is promising that the proposed method outperformed the modified CROWN-IBP ($\beta=1$) and IBP, and there seems to be a significant margin. But the proposed method fails to make a significant improvement compared to the original CROWN-IBP (the 1->0 one), e.g., the improvement on CIFAR-10 eps=8/255 or 16/255 is negligible.

Overall, I am keeping my recommendation as rejection for the current version of the manuscript.

---

> ### Author Response · Authors · 2020-11-16
> **Thank you for the valuable feedback. We will provide further detailed comments within a few days. (part1)**
>
> We will be updating comments for the other feedback soon (C3-1, C3-2, C3-5). It requires some additional experiments and we are working on it. Thank you for pointing this out.
> - C3-3: As the paper mentions that CROWN-IBP does not penalize unstable ReLU neurons with (|u|<=|l|), but how about just make the lower bound equal to the upper bound as Fast-lin (Weng et al., 2018, [https://arxiv.org/abs/1804.09699](https://arxiv.org/abs/1804.09699)) does, so the lower bound is non-zero for unstable neurons?
>     - A3-3: With the lower bound equal to the upper bound ($\underline{a}=\frac{u^+}{u^+-l^-}=\overline{a})$ as Fast-lin, it yields the same as "**CAP**-IBP" bound in Fig3 which has a looser upper bound than OURS and CROWN-IBP. It may have better smoothness, but it has worse tightness. We will check on that.
> - C3-4: An improvement over CROWN-IBP(1->1) is shown in the paper. However, the proposed method does not make enough improvement over CROWN-IBP(1->0). The improvement on CIFAR10 with eps=8/255 or 16/255 is very small (error 69.70 v.s. 69.97, 77.87 v.s. 77.88), while there is a large performance drop on MNIST with eps=0.3/0.4 (increased error 7.07->9.79 and 12.35->15.42). What will the performance of the proposed method look like if a changing  is β used in training?
>     - A3-4: Table 3 in Section J shows the results with different $\beta$-schedulings. It didn't improve performance. We will add the results on MNIST.
>     - cf. A2-1: Compared to CROWN-IBP$\_{1→0}$, OURS shows only marginal improvement or even worse in the case of $\epsilon=0.3, 0.4$ on MNIST. There is some explanation of why this $\beta$-scheduling (1→0) in (3) (from CROWN-IBP$\_{1→1}$ to IBP) can improve the performance [CROWN-IBP]. We can add on it with our understanding that CROWN-IBP$\_{1→0}$ starts with a tighter bound ($\beta=1$, CROWN-IBP only) but not overfits to small perturbation by gradually introducing the IBP objective which has a smoother landscape. We note that IBP has the most smooth landscape compared to others (Theorem1) because IBP has a relaxed gradient approximation $g=0$. We will update the paper with the above discussion.
>
> [CROWN-IBP] Zhang et al. 2019 (https://arxiv.org/abs/1906.06316)

---

> ### Author Response · Authors · 2020-11-20
> **A new link between the motivation and the proposed method (part2, end).**
>
> Thank you for the valuable feedback.
>
> - C3-1: The link between the motivation and the proposed method is not well justified. Paragraph “Favorable Landscape” somewhat explains the connection between optimizing the lower bound in relaxation and improving loss landscape. But it states that the benefit comes from preferring dead ReLU neurons to unstable ones. If the proposed method is actually trying to make more unstable neurons dead, the paper does not discuss or compare with paper Xiao et al., 2018 ([https://arxiv.org/pdf/1809.03008.pdf](https://arxiv.org/pdf/1809.03008.pdf)) which directly adds a regularization to induce ReLU stability.
>     - A3-1: In the revised version, we updated the explanation on how the proposed method can improve not only the tightness of the upper bound on the worst-case loss but also the smoothness of the classifier. We have added extensive experiments to provide a new link between the motivation and the proposed method (see Section 5 and Section E in the revised version).
>     - We investigate variants of CROWN-IBP with different $\underline{a}$ settings for unstable ReLUs, including CAP-IBP suggested in C3-3. For each setting, we sample $\underline{a}\in\{0,1\}$ with different $(p,q)$ with $P(\underline{a}=1\mid|l|>|u|)=p$ and $P(\underline{a}=1\mid|l|\leq|u|)=q$ for each neuron with pre-activation bounds $l$ and $u$. Figure 3 in the revised version shows that it tends to have more unstable ReLUs as the number of $\underline{a}$ satisfying $\underline{a}=1$ increases. This observation implies that  it is required to have smaller portion of $\underline{a}$ with $\underline{a}=1$ to have a more favorable landscape.
>     - Interestingly, CAP-IBP has a looser bound than CROWN-IBP$\_{1\rightarrow 1}$, but it has more smooth landscape as expected. As a result, CAP-IBP outperforms CROWN-IBP$\_{1\rightarrow 1}$ (VE: 74.29<75.72). This also support our claim that smoothness of the loss landscape is an important factor for training certifiably robust models.
>     - Thank you for the suggestion of ReLU Stability (RS). RS uses its regularization on adversarial training, and thus it is much less effective than other certified defenses. We compare the standard/verified error as follows:
>         - **OURS (1.09/2.28)** / RS (1.32/5.67) for $\epsilon=0.1$ on MNIST
>         - **OURS (2.42/7.84)** / RS (2.67/19.32) for $\epsilon=0.3$ on MNIST
>         - **OURS (31.49/49.42)** / RS (38.88/54.07) for $\epsilon=2/255$ on CIFAR10
>         - **OURS (56.01/69.70)** / RS (59.55/79.73) for $\epsilon=8/255$ on CIFAR10
>     - Note that we report the best (reported) performance for both methods. We updated our manuscript with the above results in Section J. We note that RS uses the MILP exact verifier [MILP] while OURS use its own relaxed verifier, but still OURS outperforms RS.
>     - In addition, we have run CROWN-IBP$\_{1\rightarrow 1}$ and our method with RS regularization. We observed that RS regularization has negligible effects on the proposed method and CROWN-IBP$\_{1\rightarrow 1}$. This is because certified defense methods already have small number of unstable ReLUs even for CROWN-IBP$\_{1\rightarrow 1}$ (<3%) which generally has the most.
> - C3-2: It does not seem to make sense that the proposed method can prefer to have dead neurons via optimizing the relaxation for unstable neurons. With such an optimization, bounds of unstable neurons become tighter, so how can such tighter bounds make the model favor unstable neurons less than CROWN-IBP with relatively looser bounds?
>     - cf) A3-1.
>     - A3-2: It is quite interesting. Some certifiable training methods with a looser bound tend to favor stable neurons. For example, IBP starts with small number of unstable ReLUs while CAP starts with large number of ReLUs as shown in Figure 2 (bottom). However, we found that a tighter bound does not always lead to many unstable ReLUs.
>     - Notation $(p/q)$ denotes the variant of CROWN-IBP with sampling $\underline{a}\in\{0,1\}$ with $P(\underline{a}=1\mid|l|>|u|)=p$ and $P(\underline{a}=1\mid|l|\leq|u|)=q$ for unstable ReLUs. For example, CROWN-IBP is equivalent to $(0/1)$ as it uses $\underline{a}=0$ for $|l|>|u|$ and $\underline{a}=1$ for $|l|\leq|u|$. We found that $(0.5/1)$ and $(1/1)$ have looser bounds than CROWN-IBP but they have more unstable ReLUs. On the other hand, $(0/0)$, $(0/0.25)$, and $(0/0.5)$ have looser bounds than CROWN-IBP and they have less unstable ReLUs. Therefore, this observation implies that it is important to have less $\underline{a}=1$ to have a more smooth landscape.
>     - We demonstrated that it is possible to achieve a tighter bound without compromising the smoothness of classifier in Figure 1 (Left).
> - C3-5: This paper missed some previous works, e.g., Xiao et al., 2018 and Weng et al., 2018 as mentioned above. More discussions or comparisons with them may be needed.
>     - A3-5: Thank you. As mentioned above, we updated the paper (Section 2, 6 and K) accordingly.

---

> > ### Comment · AnonReviewer1 · 2020-11-23
> > **Follow-up questions on the revised manuscript**
> >
> > Thanks for the response! Currently the “More favorable landscape” paragraph in Sec. 5 seems to make better sense to me. But I have some follow-up questions:
> >
> > 1. In this paper, the reported verified error of CROWN-IBP on CIFAR-10 8/255 is 69.97. I wonder if there is any difference between the setting in this paper and CROWN-IBP’s original setting? From what I know, the best result reported in CROWN-IBP is 66.94.
> >
> > 2. It seems unclear to me whether the number of unstable neurons is an important factor. On one hand, the authors argued “This is because certified defense methods already have small number of unstable ReLUs” when they find the ReLU Stability (RS) by Xiao et al. does not make an improvement here, and on the other hand, the authors shows in the “More favorable landscape” paragraph in Sec. 5 that $\underline{\alpha}$ affects the number of unstable ReLU neurons which seems to affect the training performance. These two appear to be contradictory. So could the authors resolve this contradiction and elaborate on whether they actually believe unstable neurons are important or not important?
> >
> > 3. Could the authors give some explanation on why reducing $\underline{\alpha}=1$ can improve smoothness, beyond empirical observations?
> >
> > 4. As I see from Fig 3, the numbers of unstable neurons at the end of training look very similar for “ours”, and CAP-IBP. So I wonder if the number of unstable neurons is really reduced in the proposed method, compared to CAP-IBP. By the way, I think the presentation of Fig 3 needs some improvement on the color usage, because the colors for 0/0, 0/1, and CAP-IBP respectively are too close.
> >
> > 5. By optimizing the lower bound of the relaxation, the smoothness and the tightness are both changed. So it becomes questionable whether the improvement really comes from smoothness, i.e., a better loss landscape, or the tightness instead. Could the authors provide more justifications?
> >
> > Please kindly let me know if I missed some points in the paper. Thanks!

---

> > > ### Author Response · Authors · 2020-11-23
> > > **Answers for some Follow-up questions (part1)**
> > >
> > > Thanks for the feedback! We are happy to be able to improve the paper by reflecting the feedback. We will answer some of the comments first as follows:
> > > - C1'-1: In this paper, the reported verified error of CROWN-IBP on CIFAR-10 8/255 is 69.97. I wonder if there is any difference between the setting in this paper and CROWN-IBP’s original setting? From what I know, the best result reported in CROWN-IBP is 66.94.
> > >     - A1'-1: The result 66.94 (in Table C in [CROWN-IBP]) is for the model trained with **32 TPU** cores (3200 epochs and batch size of 1024) (see Section C in [CROWN-IBP]), which we couldn't use. We used **a single Titan X (Pascal)** (400 epochs and batch size of 128). We used the DM-Medium model (=M) in Table A,B in [CROWN-IBP], for which you can refer to the results with the statistics (min, med, max) for 8 medium models including M in Table D in [CROWN-IBP]. The best results in Table D in [CROWN-IBP] is **70.51** (for 8 medium models including M), and with our reimplementation with different hyperparameters, we achieved a better result **69.97** in Table 2. Also, for IBP, they have 72.18 (for 8 medium models including M) in Table D in [CROWN-IBP], and 70.95 in Table 1 in our paper.
> > > - C1'-2: It seems unclear to me whether the number of unstable neurons is an important factor. On one hand, the authors argued “This is because certified defense methods already have small number of unstable ReLUs” when they find the ReLU Stability (RS) by Xiao et al. does not make an improvement here, and on the other hand, the authors shows in the “More favorable landscape” paragraph in Sec. 5 that α_ affects the number of unstable ReLU neurons which seems to affect the training performance. These two appear to be contradictory. So could the authors resolve this contradiction and elaborate on whether they actually believe unstable neurons are important or not important?
> > >     - A1'-2: We clarify our previous comment. We observed that **"commonly used"** RS regularization has negligible effects on **"reducing the number of unstable ReLUs"** for the proposed method and CROWN-IBP, and thus on improving their performance. We used the RS loss weight $\lambda= 0.003$ larger than the value [RS] used (0.002), but it has no effect on reducing unstable ReLUs and thus we tried another $\lambda=0.01$, which also didn't work. Note that [COLT] uses $\lambda= 0.003, 0.005$.
> > > Figure 2 in [RS] shows that RS with $\lambda=0.003$ reduces the ratio of unstable ReLUs to $20/278\approx 7$% (MNIST 0.1). However, the certified defenses we considered tend to keep the ratio less than 2% (MNIST 0.1, not in the paper) even without RS. Therefore, to see a notable effect, we must use a larger $\lambda$. After the revision, to further reduce the ratio, we are running with $\lambda=0.1,0.5,1,10,1000$. We will reply again when the results come out.
> > > - C1'-4: As I see from Fig 3, the numbers of unstable neurons at the end of training look very similar for “ours”, and CAP-IBP. So I wonder if the number of unstable neurons is really reduced in the proposed method, compared to CAP-IBP. By the way, I think the presentation of Fig 3 needs some improvement on the color usage, because the colors for 0/0, 0/1, and CAP-IBP respectively are too close.
> > >     - A1'-4: We have changed the colors from 'navy/purple/black' to 'orange/purple/lawngreen', respectively. Thank you for the comment!
> > >     - We claim the smoothness of loss landscape is important. And the number of unstable ReLUs is one of the most important factors to determine the smoothness. This is because the RGA (relaxed gradient approximation) becomes more smooth as the model becomes more linear (with less unstable ReLUs) (Theorem 1). Note that for a linear network $s(x)=Wx+b$, we have the RGA $g^{(m)}=W_{m,:}$ which yields $\nabla_{W_{m,:}}g^{(m)}=I$, and thus $g^{(m)}$ is very smooth.
> > >     - However, the number of unstable ReLUs alone cannot fully explain the smoothness. The number of unstable ReLUs in OURS seems similar to CAP-IBP, but the gradient difference (left-hand side in (4)) does not. The gradient difference is more directly related to the smoothness of loss landscape. We added Figure 18 in Section Q, which shows CAP-IBP has a less smooth landscape than OURS, but it is more smooth than CROWN-IBP ($\beta=1$). It is also consistent with the loss variation plot, Figure 7 in Section E. This leads to a performance of CAP-IBP between OURS and CROWN-IBP ($\beta=1$) (71.45 (OURS) <74.29 (CAP-IBP)<75.72 (CROWN-IBP ($\beta=1$)). Please also kindly check Figure 10.
> > >
> > > We will answer for the other comments in 24 hours. Thank you again for your valuable comments.
> > >
> > > [CROWN-IBP] Zhang et al. 2019 ([https://arxiv.org/abs/1906.06316](https://arxiv.org/abs/1906.06316))
> > >
> > > [RS] Xiao et al., 2018 ([https://arxiv.org/pdf/1809.03008.pdf](https://arxiv.org/pdf/1809.03008.pdf))
> > >
> > > [COLT] Balunovic and Vechev, 2019 (https://openreview.net/forum?id=SJxSDxrKDr)

---

> > > > ### Author Response · Authors · 2020-11-24
> > > > **Answers for some Follow-up questions (part2)**
> > > >
> > > > - C1-2': It seems unclear to me whether the number of unstable neurons is an important factor. On one hand, the authors argued “This is because certified defense methods already have small number of unstable ReLUs” when they find the ReLU Stability (RS) by Xiao et al. does not make an improvement here, and on the other hand, the authors shows in the “More favorable landscape” paragraph in Sec. 5 that α_ affects the number of unstable ReLU neurons which seems to affect the training performance. These two appear to be contradictory. So could the authors resolve this contradiction and elaborate on whether they actually believe unstable neurons are important or not important?
> > > >     - cf. A1'-2
> > > >     - A1'-2-2:  To see a notable effect of RS, we have run CROWN-IBP ($\beta=1$) with $\lambda=0.1,0.5,1,10,1000$. We found that $\lambda< 0.5$ has no effect on reducing the number of unstable ReLUs, and thus not on improving the smoothness. On the other hand, with $\lambda\geq0.5$, we observed that RS reduces the number of unstable ReLUs. RS with $\lambda=0.5,1,10$ also lead to a better loss variation and gradient difference. This supports our claims that unstable ReLUs are closely related to the smoothness of the loss landscape. However, as [RS] mentioned "placing too much weight on RS Loss can decrease the model capacity, potentially lowering the provable adversarial accuracy", the models trained with $\lambda\geq1$ couldn't obtain the same level of tightness as CROWN-IBP ($\beta=1$) and a significant improvement on robustness. Please kindly check Section R in the revised version.
> > > > - C1'-3: Could the authors give some explanation on why reducing α_=1 can improve smoothness, beyond empirical observations?
> > > >     - A1'-3: We propose the following hypothesis based on our observations:
> > > >     - As Section C.1 shows, when $W^{(k)}\_{i,j}<0$, we choose the lower linear bound $\underline{h}\_j^{(k-1)}$. If the lower linear bound has a slope $\underline{a}\_j^{(k-1)} =1$, the objective encourages the model to minimize the maximum of $W^{(k)}\_{i,j}\underline{h}\_j^{(k-1)}(x)$ for $x\in [l,u]$. Since it attains the maximum at $x=l$, it minimizes $W^{(k)}\_{i,j}\underline{a}\_j^{(k-1)}l=W^{(k)}\_{i,j}l$, which leads to maximize $l$. On the other hand, with a slope $\underline{a}\_j^{(k-1)} =0$, there is no such effect. We observed that in case of CROWN-IBP ($\beta=1$) which has more $\underline{a}=1$, it encourages the model to increase $l$. Therefore, it has tendency to move the dead ReLUs to unstable ReLUs, and unstable ReLUs to active ReLUs. Since number of dead ReLUs outnumber the others, more dead ReLUs become unstable than the number of unstable ReLUs become active.
> > > >     - We empirically supported this hypothesis. We investigated the change of neurons in the first ReLU layer on a fixed example $x$ during the early ramp-up period trained by CROWN-IBP ($\beta=1$) and OURS. In the first 40 epochs of CROWN-IBP ($\beta=1$), 3802 dead neurons have changed to unstable ReLUs while 3351 unstable ReLUs has moved to active ReLUs. On the other hand, OURS in the same period, 2330 dead ReLUs has been changed to unstable ReLUs and 2220 unstable ReLUs has moved to active ReLUs. As a result, CROWN-IBP ($\beta=1$) tends to have more unstable ReLUs than OURS in the ramp-up phase as shown in Figure 15 in Section J.
> > > > - C1'-5: By optimizing the lower bound of the relaxation, the smoothness and the tightness are both changed. So it becomes questionable whether the improvement really comes from smoothness, i.e., a better loss landscape, or the tightness instead. Could the authors provide more justifications?
> > > >     - During training, we are given a weight $\theta$ for each weight update. Near $\theta$, we can measure a "local" smoothness of the loss landscape. Therefore, we can measure the local smoothness throughout the training phase as in Figure 1 (Left) and Figure 10. We use the term "smoothness" for the behavior during the overall training phase.
> > > >     - For a small $\epsilon$, CAP outperforms others since CAP utilizes tighter bounds. As mentioned in A2-2, tightness seems more important than smoothness for a small $\epsilon$. On the other hand, as shown in Section Q, CAP-IBP (or IBP) has looser bounds than CROWN-IBP but it has a relatively more smooth loss landscape, and it leads to better performance for a large $\epsilon$. In this case, smoothness plays an important role in improving performance.
> > > >     - Our method has both better tightness and smoothness than CROWN-IBP ($\beta=1$). By achieving the best of both worlds, the tightness of the proposed method has improved the performance for a small $\epsilon$, while the smoothness of the proposed method helps the optimization process, which also leads to better performance for a large $\epsilon$. To conclude, the proposed method can achieve a decent performance under a wide range of perturbations as shown in Table 1.

---

### Official Review · AnonReviewer2 · 2020-10-28
**Central claims unsupported by evaluation**

**Rating:** 3
**Confidence:** 5

**Review:**

This paper claims to make three contributions:  (i) a theoretical and experimental demonstration that the driving force behind performance in certifiable training methods is loss landscape smoothness (ii) a new training method and domain designed to have a smooth loss landscape (iii) an evaluation showing their new training method performs comparably to the state of the art.


Pros:
* The paper provides some experimental analysis support a hypothesis that has been conjectured since Mirman et al. (2018), that smooth loss landscapes produced by smoother transformers (zSmooth/hSmooth) would improve training.
* The paper does show experimentally that certain methods tend to have less smooth loss landscapes.
* The theoretical result connects the smoothness of the relaxed gradient approximations to the smoothness of the loss landscape, which may be useful for designing future certifiable training methods.

Cons:
* In terms of accuracy and verified error, the proposed system performs comparably to some pre-existing systems.  The main claim that the authors have identified an important factor for improving certified training remains unsupported by the inability to improve noticeably upon results from prior work.
* The authors do not compare it to the state of the art, COLT [1], which achieves for example 21.6/39.5 (standard, verified error) compared to this paper’s 31.49/49.42 results on CIFAR10 2/255.  This is a much more significant difference than for any of the comparisons made to other work in this paper.
* It is unclear how fast the proposed method is.  No comparison appears to be made on training speed or memory usage compared with other systems.  Given comparable accuracy and verified error results, I would not see a reason to use this system if this one is slower.
* I am not sure about the utility of the experiment on tightness and Figure 3.  Different methods could have different scales of loss while producing similarly verifiable networks (as this paper in fact demonstrates).  Further, even the worst case margin for a class could be fooled by a network with entirely  zero weights.
* Loss value is also compared between methods in Figure 1.
* The paper claims that state of the art methods such as CROWN-IBP and CAP are held back by their non-smooth loss landscapes while failing to meaningfully outperform them.  5 out of 10 of the standard accuracies are worse than CROWN-IBP’s for example.
* Section 4.1 claims that the experiments show that “the non-smoothness of the relaxed gradient approximation of linear relaxations negatively affects their performance” yet the evidence presented is circumstantial, and not capable of indicating causality.
* Section 4.2 claims that the theoretical analysis states that some landscapes are more favourable  than others, whereas in fact it only demonstrates the smoothness of some landscapes not their “favourability.”
* The paper leaves out additional related work.  In particular, I would like to see a discussion of the difference between the analysis used by Fastened Crown [2].

While this paper does provide a new technique with some theoretical justification, unfortunately neither the theoretical justification or experimental results are significant enough to recommend acceptance.  Furthermore, because the paper’s central assumption, that a smooth loss landscape leads to better results, is unsupported by their own experimental results, I must rate the paper a rejection.

[1] Mislav Balunovic and Martin Vechev. Adversarial training and provable defenses: Bridging the gap. In International Conference on Learning Representations, 2020.
[2] Lyu, Z., Ko, C.-Y., Kong, Z., Wong, N., Lin, D., and Daniel, L. Fastened crown: Tightened neural network robustness certificates.  In AAAI, 2020.

=========================================================================================
Update:

While some of my points have been addressed and the quality of the paper has been improved, my main concern, that the experiments do not support the central claim.

The authors argue that because their method has a smoother loss landscape then similar methods, it performs better.  In the updated paper, the evidence that the method performs better than similar methods has been made clearer, but the improvement is is still marginal.  The more pressing matter however is that the conclusion that the cause of the improvement is from a change in the loss landscape smoothness is based only on a qualitative comparison of only five methods (as the benefit is not consistent between methods in any category).  This is enough to at best demonstrate a weak correlation, but not enough to demonstrate causation.

---

> ### Author Response · Authors · 2020-11-15
> **Central claims supported by comparison with IBP, CAP, and CROWN-IBP$_{1\rightarrow 1}$ (part1)**
>
> Thank you for the valuable feedback. We will provide further detailed comments within a few days.
> - C2-1: In terms of accuracy and verified error, the proposed system performs comparably to some pre-existing systems. The main claim that the authors have identified an important factor for improving certified training remains unsupported by the inability to improve noticeably upon results from prior work.
>     - A2-1: The main purpose of this paper is to understand previous linear relaxation-based methods (IBP,  CAP, and CROWN-IBP$\_{1\rightarrow 1}$ ) in terms of tightness of bound and smoothness of landscape and to propose a training method that can utilize both terms. In the analysis (Section 4), we compared the three methods (IBP, CROWN-IBP$\_{1\rightarrow 1}$, CAP) with OURS. Note that CROWN-IBP$\_{1\rightarrow 0}$ is not compared because of the complex behavior of $\beta$-scheduling. In Table 1, OURS shows the best performance among those 4 (except CROWN-IBP$\_{1\rightarrow 0}$) except for the case when $\epsilon$ is small. When $\epsilon$ is small, OURS is the second best (see A2-2 for details).
>     - However, compared to CROWN-IBP$\_{1→0}$, OURS shows only marginal improvement or even worse in the case of $\epsilon=0.3, 0.4$ on MNIST. There is some explanation of why this $\beta$-scheduling (1→0) in (3) (from CROWN-IBP$\_{1→1}$ to IBP) can improve the performance [CROWN-IBP]. We can add on it with our understanding that CROWN-IBP$\_{1→0}$ **starts with a tighter bound** ($\beta=1$, CROWN-IBP only) **but not overfits to small perturbation by gradually introducing the IBP objective which has a smoother landscape.** We note that IBP has the most smooth landscape compared to others (Theorem1) because IBP has a relaxed gradient approximation $g=0$. We will update the paper with the above discussion.
>
> - C2-2: The authors do not compare it to the state of the art, COLT [1], which achieves for example 21.6/39.5 (standard, verified error) compared to this paper’s 31.49/49.42 results on CIFAR10 2/255. This is a much more significant difference than for any of the comparisons made to other work in this paper.
>     - cf) A2-1
>     - A2-2: As mentioned above, OURS shows the best performance among the four methods (except CROWN-IBP$\_{1→0}$) except for the case when $\epsilon$ is small. Especially, under $\epsilon=2/255$ on CIFAR-10 or $\epsilon=0.01$ on SVHN, CAP is better than OURS. This can be explained by the fact that CAP has a tighter bound than OURS (see Fig 1, Left, at epoch=10). Similarly, COLT shows better results than OURS for $\epsilon=2/255$ on CIFAR-10. But, this does not contradict our understanding that smoothness and "tightness" are important. For small $\epsilon$ values, tightness seems more important than smoothness. Our understanding is that with a non-smooth landscape (e.g. CAP, CROWN-IBP$_{1→1}$), it tends to overfit to a small perturbation case during $\epsilon$-scheduling. Thus, it does not generalize well to a large perturbation. Therefore, when trained with a small (target) perturbation, it shows decent robustness against the perturbation. Note that both CAP and COLT have worse performance than OURS for large $\epsilon$ since they overfit to small $\epsilon$ during $\epsilon$-scheduling. We summarize the VE (verification error) as follows:
>     - **OURS (2.36)** / CAP (3.19) / COLT (2.9) for $\epsilon=0.1$ on MNIST
>     - **OURS (9.79)** / CAP (47.85) / COLT (14.3) for $\epsilon=0.3$ on MNIST
>     - OURS (49.42) / CAP (48.50) / **COLT (39.5)** for $\epsilon=2/255$ on CIFAR10
>     - **OURS (69.70)** / CAP (73.02) / COLT (72.5) for $\epsilon=8/255$ on CIFAR10
>     - We also note that COLT additionally uses MILP and [Ehlers] to compute the VE (which takes roughly 2 days for CIFAR-10 test data), and thus it might be unfair to compare with this value directly. We will update the paper with the above discussion with COLT.
>
> - C2-3: It is unclear how fast the proposed method is. No comparison appears to be made on training speed or memory usage compared with other systems. Given comparable accuracy and verified error results, I would not see a reason to use this system if this one is slower.
>     - A2-3: We will update the complete comparison results soon. Our method is not as fast as IBP and CROWN-IBP (x1/2), but faster than CAP (x6-7) and COLT. We note the approximate speedup in parentheses. In the case of COLT, it is hard to compare the speed since they only use a 4-layer small network (we use 7-layer) and they require 4 stages (one stage for each layer) of training with 200 epochs each. Anyhow, COLT is slower than CAP. In addition, our method requires more memory than IBP and CROWN-IBP but much less than CAP and COLT. Again, we emphasize that our focus is on understanding certifiable training, and thus speed and memory efficiency are not our focus unless it does not run.

---

> ### Author Response · Authors · 2020-11-15
> **tightness, smoothness and favourability (part2)**
>
> - C2-4: I am not sure about the utility of the experiment on tightness and Figure 3. Different methods could have different scales of loss while producing similarly verifiable networks (as this paper in fact demonstrates). Further, even the worst-case margin for a class could be fooled by a network with entirely zero weights.
>     - A2-4: In Fig 3 and Fig 13 in Section M, we compared the verification loss for each verification method. We emphasize that for each figure, the model to be evaluated is fixed (e.g. Fig 3=OURS and (a) in Fig 13=CROWN-IBP$\_{1\rightarrow0}$), but it is evaluated with different evaluation methods. Fig 3 shows that our objective has a tighter upper bound $\mathcal{L}(\overline{s}(x),y)$ on the worst-case loss over valid perturbations defined in (1) than other choices of $\underline{a}$. Note that the other methods use the same form of the objective, but they differ in how to compute the worst-case logit. The objective loss can be understood as a negative log-likelihood for the worst-case logit (margin score) $\overline{s}(x)$ to be classified as $y$. Thus, higher loss indicates a higher verification error (we also empirically observed this result). We will add the verification error value for each violin plot. Moreover, Fig 1 ($\epsilon=8/255$) ends up the loss with the order of OURS<IBP<CAP<CROWN-IBP$\_{1\rightarrow 1}$ and the results in Tab1 shows the exact same order, OURS (69.70)<IBP (70.95)<CAP (73.02)<CROWN-IBP$\_{1\rightarrow 1}$ (75.37). We don't fully understand what you mean by the last sentence. With an entirely zero weight network, the worst-case margin should be 0 and it would yield -ln(0.1)=2.3 loss. Then the four evaluation has no difference. However, we aim to demonstrate that compared to other choices of lower linear bound in the relaxation, our training objective is a tighter upper bound on worst-case loss during training. To this end, the losses are evaluated on a trained model, not a random network.
> - C2-5: Loss value is also compared between methods in Fig 1.
>     - A2-5: As mentioned in A2-4, we observed that higher loss in Fig 1 directly indicates higher verification error. For example, for the input $x$ with label $y=0$, let's say, the worst-case logit is $\overline{z}_A=[1.5,0.5]$ with model A and $\overline{z}_B=[0.5,1.0]$ with model B, or equivalently, the worst-case margin score is $\overline{s}_A=[0,-1.0]$ with model A and $\overline{s}_B=[0,0.5]$ with model B. Model A provides a better bound with lower loss/error than model B.
> - C2-6: The paper claims that state of the art methods such as CROWN-IBP and CAP are held back by their non-smooth loss landscapes while failing to meaningfully outperform them. 5 out of 10 of the standard accuracies are worse than CROWN-IBP’s for example.
>     - A2-6: We claim that CROWN-IBP$\_{1→1}$ (not CROWN-IBP$\_{1→0}$) is held back by their non-smooth loss landscapes and OURS outperforms CROWN-IBP$\_{1→1}$ for 9 out of 10 (except $\epsilon=0.1$ on MNIST) both in standard and robust error. Since CROWN-IBP$\_{1→0}$ utilizes $\beta$-scheduling during training, its loss landscape is a mixture of CROWN-IBP$\_{1→1}$ and IBP and thus it is hard to interpret the effects (cf. A2-1). See A2-2 in the case of CAP.
> - C2-7: Section 4.1 claims that the experiments show that “the non-smoothness of the relaxed gradient approximation of linear relaxations negatively affects their performance” yet the evidence presented is circumstantial, and not capable of indicating causality.
>     - A2-7: We try to link the relationship between the loss objective of certifiable training methods and their performance as follows: smoothness of the relaxed gradient approximation → (Thm1 + Fig6 in Section G) → smoothness of loss landscape → (A2-8) → better training without getting stuck in a local minimum → better performance.
> - C2-8: Section 4.2 claims that the theoretical analysis states that some landscapes are more favourable than others, whereas in fact it only demonstrates the smoothness of some landscapes not their “favourability.”
>     - A2-8: The smoothness of the loss landscape in the parameter space is quite a natural factor to have favourability to learn the optimal parameter by gradient descent method. For example, certain network designs such as skip connection and batch-normalization smooth the loss function and make the network easier to be trained [Visualizing, BN]. We also refer the readers to a recent (NeurIPS 2020) paper [PAS] related to such topic on adversarial training and its loss landscape. They also consider the difference of the loss gradients as in Thm1 and Fig 6 in Section G.
>     - For a direct observation of favourability, see Fig 5 in Section F that draws the learning curves for the target perturbation. It demonstrates IBP and OURS has high favourability in learning robustness to the target perturbation.

---

> ### Author Response · Authors · 2020-11-15
> **Related Work (part3, end)**
>
> - C2-9: The paper leaves out additional related work. In particular, I would like to see a discussion of the difference between the analysis used by Fastened Crown [2].
>     - A2-9: There are many other related works on "certified evaluation (certification)" such as [Fast-Lin], [CROWN], and [CNN-Cert]. However, our work focuses on "certifiable training". We will update them in the Related Work Section.
>
> Note that the notation "Figure n" points to "Figure n in the original paper". After resubmission, we will edit them accordingly.
>
> [CROWN-IBP] Zhang et al. 2019 ([https://arxiv.org/abs/1906.06316](https://arxiv.org/abs/1906.06316))
>
> [Fast-Lin] WEng et al. 2018 ([https://arxiv.org/abs/1804.09699](https://arxiv.org/abs/1804.09699))
>
> [CROWN] Zhang et al. 2018 ([https://arxiv.org/abs/1811.00866](https://arxiv.org/abs/1811.00866))
>
> [CNN-Cert] Boopathy et al. 2018 ([https://arxiv.org/abs/1811.12395](https://arxiv.org/abs/1811.12395))
>
> [Ehlers] Ehlers et al. 2017 ([https://arxiv.org/abs/1705.01320](https://arxiv.org/abs/1705.01320))
>
> [PAS] Liu et al. 2020 ([https://arxiv.org/abs/2006.08403](https://arxiv.org/abs/2006.08403))
>
> [Visulalizing] Li et al. 2018 ([https://arxiv.org/pdf/1712.09913.pdf](https://arxiv.org/pdf/1712.09913.pdf))
>
> [BN] Santukar et al. 2019 ([https://arxiv.org/abs/1805.11604](https://arxiv.org/abs/1805.11604))

---

### Official Review · AnonReviewer3 · 2020-10-29
**Good paper with interesting findings**

**Rating:** 7
**Confidence:** 3

**Review:**

In this paper, the authors studied the role of loss landscape in training certifiable robust models. The authors reviewed linear relaxation based methods, and showed that Interval Bound Propagation (IBP) is a special case of linear relaxation based methods. Although linear relaxation based methods have a tighter bound on worst case loss with adversarial perturbations than IBP based method, the authors found in numerical studies that towards the end of training, IBP outperforms linear relaxation based methods. The authors hypothesized that this was because IBP loss landscape was more smooth, which helped optimization. The authors demonstrated in a theorem that IBP loss was indeed more smooth under certain assumptions. Based on this insight, the authors proposed a favorable landscape method. The authors showed in numerical studies that the sum over the worst-case margin for each class is lowest for their method. The loss of their method is also the most smooth among competing methods. Their method achieved a consistent performance in a range of perturbations, which is not achieved in competing methods.

The paper is clearly written. The insight is interesting, and could help future researchers.

On page 4, in Figure 1, what is the mathematical definition of "variation"? Why does the variation of CROWN-IBP first increase, and then decrease? A related question is why does the proportion of unstable ReLu first increase and then decrease in CROWN-IBP? Does it have something to do with the curvature shown in Figure 1 (right)?

One page 7, it would be great to also show the comparison of training time between these methods.

---

> ### Author Response · Authors · 2020-11-14
> **Thank you for the valuable feedback. We will provide further detailed comments within a few days.**
>
> Q1-1: On page 4, in Figure 1, what is the mathematical definition of "variation"?
> - A1-1: The definition of variation is defined in (8) in Appendix B as follows:
> $\mathcal{L}(\overline{s}(\theta(t))) \text{ where } \mathcal{L}(\overline{s}(\theta))\equiv\mathcal{L}(\overline{s}(x,y,\epsilon;\theta),y) \text{ and }\theta(t)\equiv \theta_0-t\eta\nabla_\theta \mathcal{L}(\overline{s}(\theta_0)) \text{ for } t\in[0,5]$
>  It is the range of the loss values along the gradient descent direction with the stepsize $\in[0,5]\times\eta$ where $\eta$ is the learning rate.
>
> Q1-2: Why does the variation of CROWN-IBP first increase, and then decrease?
>
> - A1-2: It is hard to tell why they show such behavior. Certifiable training prefers stable (dead/alive) ReLUs (97-99\%) to unstable ReLUs (1-3\%) since unstable ReLUs make the upper bound looser (see Fig 11) because of the relaxation in unstable ReLU. However, without unstable ReLU, the model is (*provably/certifiably*) linear under the perturbation and has very low expressive power. Therefore, the two factors conflict with each other. We hypothesize that as CROWN-IBP$_{1\rightarrow1}$ approaches to a flatter local minimum, the model is stabilized and focuses on reducing unstable ReLUs. This behavior is likely to be related to the curvature in Figure 1 (right). Near a flatter minimum, the curvature becomes flattened and loss variations are reduced.
>
> Q1-3: A related question is why does the proportion of unstable ReLu first increase and then decrease in CROWN-IBP?
>
> - cf) A1-2
>
> Q1-4: Does it have something to do with the curvature shown in Figure 1 (right)?
>
> - cf) A1-2
>
> Q1-5: One page 7, it would be great to also show the comparison of training time between these methods.
>
> - A1-5: Roughly speaking, our method is not as fast as IBP and CROWN-IBP (x1/2), but faster than CAP (x6-7). We note the approximate speedup in parentheses. We will update complete comparison results soon.

---

### Author Response · Authors · 2020-11-20
**Summary of Revision (1)**

Dear reviewers,
Thanks for the valuable comments. We have revised our paper and please kindly check it. We hope we have addressed all concerns raised by the reviewers. We highlighted the changes in the revised version. We summarized the responses as follows: (Note that the original numbering is no longer applied)
- **[Rev2, Rev3] Speed comparison updated.**
    - We updated the paper with the following speed comparison in Section O:
    - OURS: 115.9 sec / epoch
    - CROWN-IBP$\_{1\rightarrow1}$: 51.68 sec / epoch
    - IBP: 14.85 sec / epoch
    - CAP (batch size 32, 1 GPU): 751.0 sec / epoch
    - CAP (batch size 64, 1 GPU): 724.6 sec / epoch
    - CAP (batch size 128, 2 GPUs): 387.9 sec / epoch
    - Unless otherwise stated, the times are measured on a single TITAN X (Pascal) with batch size of 128.
- **[Rev2] Our claims are supported by the experiments with IBP, CROWN-IBP $(\beta=1)$, and CAP. Tab 1 updated.**
    - We found that the smoothness of the loss landscape and the tightness of the upper bound on the worst-case loss are important factors in building certifiably robust models. IBP has a smooth loss landscape, but it has a looser bound. On the other hand, CROWN-IBP $(\beta=1)$, and CAP have relatively tighter bound, but they have a less smooth loss landscape. We proposed a certifiable training method with a smooth loss landscape and a tighter bound, and thus it achieved the best results among them. To emphasize this point, we revised Table 1 in the revised version.
    - We add an explanation of why the $\beta$-scheduling (1→0) in (3) (from CROWN-IBP to IBP) can improve the performance in CROWN-IBP$\_{1\rightarrow 0}$.
    - New paragraph "Understanding $\beta$-scheduling" added
- **[Rev1,Rev2] Comparison to prior work including RS and COLT updated.**
    - We updated the paper with the following comparison to other prior work in Section K:
    - OURS (1.09/**2.28**) / RS (1.32/5.67) / [DiffAI] (1.3/4.2) / COLT (0.8/2.9) for $\epsilon=0.1$ on MNIST
    - OURS (2.42/**7.84**) / RS (2.67/19.32) / [DiffAI] (3.4/10.7) / COLT (2.7/14.3) for $\epsilon=0.3$ on MNIST
    - OURS (31.49/49.42) / RS (38.88/54.07) / [DiffAI] (37.7/54.5) / COLT (21.6/**39.5**) for $\epsilon=2/255$ on CIFAR10
    - OURS (56.01/**69.70**) / RS (59.55/79.73) / [DiffAI] (53.8/72.8) / COLT (48.3/72.5) for $\epsilon=8/255$ on CIFAR10
- **[Rev2] Verified Losses indicate the verified error. Thus the losses over the methods are comparable.**
    - We addressed related concerns in A2-4,5,6,7,8. Please kindly check it.
    - Note that all of our main comparison methods (CAP, IBP, CROWN-IBP, OURS) have in common that they use loss function as the cross entropy between $\overline{s}(x)$ and $y$. As a result, all the loss functions are on the same scale. Since all four methods have the same scale, a higher loss in Fig1 directly indicates higher verified error.
    - We hope we have addressed the related concerns raised in C2-4,5,6,7,8.
- **[Rev1] A new link between the motivation and the proposed method.**
    - cf. A3-1,2
    - New paragraph "More favorable landscape via less $\underline{a}= 1$" added
    - We added Fig 3 and Fig 5 in Section 5, 6 in the revised version.

[DiffAI] Mirman et al., 2018 (http://proceedings.mlr.press/v80/mirman18b.html)

---

> ### Author Response · Authors · 2020-11-21
> **Revision (2)**
>
> We changed Figure 5 (legend and caption) and Appendix E (Table 3).

---

> > ### Author Response · Authors · 2020-11-23
> > **Revision (3)**
> >
> > We added Section Q (Fig 18) and changed the colors in Fig 3.

---

> > > ### Author Response · Authors · 2020-11-24
> > > **Revision (4)**
> > >
> > > We added Section R (Fig 19, 20).

---

### Author Response · Authors · 2020-11-25
**[Rev2] Additional comments on C2-4,5**

- **[Rev2] Additional comments on C2-4,5:**
    - In standard training, the standard cross-entropy loss $\mathbb{E}_{(x,y)\sim D}\mathcal{L}(z(x;\theta),y)$ is used as an objective with a function $z(\cdot;\theta)$. Each different structure (Residual layer, BatchNorm, etc.) uses different function $z(\cdot;\theta)$ and predicts with $\text{argmax}_i z_i(x;\theta)$. It is common to examine the standard cross-entropy losses for different structures to understand the effect of the structure such as Residual layer [Visualizing] and BatchNorm [BN].
    - Similarly, in certified defenses, a verified loss $\mathbb{E}_{(x,y)\sim D}\mathcal{L}(\overline{z}(x;\theta),y)$ is used as an objective with a function $\overline{z}(\cdot;\theta)$. Each different method (CAP, IBP, etc.) uses different function $\overline{z}(\cdot;\theta)$ and verifies with $\text{argmax}_i\overline{z}_i(x;\theta)$.
    - Since we use the same logit function $\overline{z}(\cdot;\theta)$ used in the certification for a certifiable training, there is no difference between our evaluation and those in standard training ([Visualizing] and [BN]).
    - From these reasons, we highly believe that loss value can be compared between methods as in Fig 1 and our analysis is valid.

[Visulalizing] Li et al. 2018 ([https://arxiv.org/pdf/1712.09913.pdf](https://arxiv.org/pdf/1712.09913.pdf))

[BN] Santukar et al. 2019 ([https://arxiv.org/abs/1805.11604](https://arxiv.org/abs/1805.11604))

---

### Author Response · Authors · 2021-12-02
**link to published version**

We thank the reviewers again. With their insightful and valuable comments, the contents and the clarity of our paper are much improved in the revised version. Please check our published version at the following link:

https://papers.nips.cc/paper/2021/hash/07c5807d0d927dcd0980f86024e5208b-Abstract.html

---

### Decision · Program_Chairs · 2021-01-07
**Final Decision**

**Decision:**

Reject

**Comment:**

The authors argue that tighter relaxations for certified robustness suffer from a worse loss landscape and thus are outperformed by the much simpler and less tight IBP relaxation and come up with a new relaxation to overcome this problem.

After the rebuttal there still remain doubts about the reasoning regarding the loss landscape (even though I acknowledge that the authors have invested significant amount of work to support their hypothesis). Moreover, the differences to existing certified training methods is small or the proposed method performs worse while being significantly more expensive (in particular if one takes into account the results which are reported on the IBP-Crown github page where the reported numbers are significantly lower than reported in the present paper) so that the benefit is unclear.

Thus the majority of the reviewers still suggests rejection and I agree with that even though I think that the paper has its merits and I encourage the authors to continue this line of work. For a next version, the authors should evaluate all the methods ideally with an exact verification method resp. use the best relaxation for all methods. Otherwise the differences can come just from the weaker relaxation but not from a difference in real robustness.